# Generalization Error Bounds for Two-stage Recommender Systems with Tree Structure

**Jin Zhang[1], Ze Liu[2], Defu Lian**[*][1, 2, 3]**, Enhong Chen**[1, 2, 3]

[1] School of Artificial Intelligence and Data Science, University of Science and Technology of China
[2] School of Computer Science and Technology, University of Science and Technology of China
[3] State Key Laboratory of Cognitive Intelligence, Hefei, Anhui, China
{jinzhang21, lz123}@mail.ustc.edu.cn, {liandefu, cheneh}@ustc.edu.cn

## Abstract

Two-stage recommender systems play a crucial role in efficiently identifying relevant items and personalizing recommendations from a vast array of options. This paper, based on an error decomposition framework, analyzes the generalization error for two-stage recommender systems with a tree structure, which consist of an efficient tree-based retriever and a more precise yet time-consuming ranker. We use the Rademacher complexity to establish the generalization upper bound for various tree-based retrievers using beam search, as well as for different ranker models under a shifted training distribution. Both theoretical insights and practical experiments on real-world datasets indicate that increasing the branches in tree-based retrievers and harmonizing distributions across stages can enhance the generalization performance of two-stage recommender systems.

## 1 Introduction

Recommender systems play a crucial role in many online services, such as e-commerce [29], digital streaming [4], and social media [2], influencing consumer behavior, media consumption, and social interaction. It needs to quickly identify a few relevant items from millions or billions of options, and personalize to the dynamic needs of large numbers of users with low response latency. A widely adopted solution to this problem is the two-stage recommender system. In the first stage, a computationally efficient retriever preselects a small number of candidates from a large pool. In the second stage, a slower but more accurate ranker narrows down and reorders these candidates before presenting them to the user. In this way, a well-balanced trade-off between efficiency and accuracy is achieved that meets the demands of real-world scenarios.

The retrievers are often heterogeneous, popular choices like matrix factorization [15, 19], two-tower [23], recurrent neural networks [3], and so on. In recent years, the tree-structured retriever model [8, 29, 28, 7], which takes advantage of the tree structure, usually combined with a greedy algorithm to identify relevant items quickly, has demonstrated commendable performance and efficiency. The ranker typically uses enriched features as input, combined with a complex model, to enhance prediction accuracy. The computational costs are generally linear relative to the number of items at deployment [4, 18].

Despite the practical success of two-stage models, particularly those based on tree structures, theoretical research in this area remains limited. To fill in the gap, we start from the perspective of generalization error to investigate the upper bounds of generalization error for these models to promote understanding of their generalization capabilities.

---

[*]Corresponding Author

38th Conference on Neural Information Processing Systems (NeurIPS 2024).

In this paper, we decompose the generalization error of two-stage models across each stage. Using Rademacher complexity as a methodological tool, our analysis encompasses a range of models prevalent in two-stage methods. This includes tree-structured retriever models employing beam search, such as linear model, multilayer perceptron, and target attention model. Besides, we give generalization error bound for ranker models under shifted training distributions. The theoretical results show that tree models with increased branches and rankers trained on harmonized distributions can improve generalization performance, and we validate these findings on real-world datasets.

To summarize, the main contributions of this work are summarized as follows:

- We are the first to analyze the learnability of tree-based retriever models in recommender systems and prove the generalization upper bound for various tree-based retrievers using beam search. Both theoretical insights and practical experiments confirm that expanding the number of branches in tree-based retrievers enhances their ability to generalize.
- We establish an error decomposition framework for analyzing generalization errors of two-stage recommender systems and theoretically derive the optimized training objectives for the ranker models within this framework.
- We prove the generalization upper bounds for different ranker models under shifted training distribution, highlighting the significant impact of disparities between training and inference data distributions on generalization errors. Theoretical and empirical findings indicate that harmonizing distributions across stages enhances the overall generalization performance of two-stage recommender systems.

The remainder of this paper is organized as follows: Section 2 provides an overview of related work, Section 3 presents the notation and background, Section 4 presents the main results and analytical techniques, and Section 5 provides the conclusion of the paper. Finally, the missing proofs and experimental settings are provided in the appendix.

## 2 Related Work

### 2.1 Two-stage Recommender Systems

Two-stage recommender systems with candidate generation followed by ranking have been widely adopted in the industry, including YouTube [4, 23, 26], Linkedin [2], Pinterest [6]. Some works focus more on improving candidate generation, particularly under tree structures. TDM [29] efficiently manages candidate retrieval in large-scale systems using a hierarchical tree-index strategy. JTM [28] improves on TDM by jointly optimizing the tree index structure and the user node preference prediction model. DeFoRec [8] extends the loss function used in TDM from a binary probability to a multi-class softmax loss. Other works, like [18], study off-policy learning for two-stage recommender systems, where the goal is to learn a good recommendation policy from the typically abundant logged data. This approach is possibly most related to our work in the ranker component. The main proposal of [18] is to modify the training objective by adding importance weights based on the ranker's probability of recommending each item. With adjustments facilitating gradient descent optimization, the authors show empirical improvements compared to a system trained without importance weighting. [13] propose a modification of naive bandit method deployment in two-stage recommenders that improves results by sharing inferred statistics between ranker and nominators with minimal computational overhead. [25] aims to provide an LLM-based two-stage recommender that uses a large language model as a ranker to improve performance.

### 2.2 Theoretical Work

In this subsection, we discuss the theoretical work related to our study, as well as other theories related to two-stage models. [1] propose a multi-class, hierarchical data-dependent bound on the generalization error of classifiers deployed in large-scale taxonomies to explain several empirical results related to the performance of hierarchical classifiers. Our analysis of tree structure models is inspired by this work. We extend it to the search method of beam search and provide a more refined estimate for the tree model. [17] investigates generalization error bounds for extreme multi-class classification with minimal dependence on the class set by using multi-class Gaussian complexity to construct bounds for multi-class problem. [12] theoretically demonstrated that nominator count

and training objectives significantly impact two-stage recommender performance and linked two-stage recommenders to Mixture-of-Experts models to show performance improvements by allowing nominators to specialize. [14] quantitatively assesses the asymptotic convergence properties of the two-tower model applied in two-stage recommenders toward an optimal recommender system.

# 3 Preliminaries

## 3.1 Notation

We use the following notational conventions: bold lowercase and uppercase letters for vectors and matrices respectively, such as $\boldsymbol{a}$ and $\boldsymbol{A}$, and non-bold letters for scalars or constants, such as $c$ and $B$. For vectors, $\|\boldsymbol{a}\|_p$ denotes the $\ell_p$ norm; we drop the subscript for the $\ell_2$ norm. For matrix,

$$\|\boldsymbol{A}\|_p = \sup_{\|\boldsymbol{x}\|_p=1} \|\boldsymbol{A}\boldsymbol{x}\|_p$$

denotes matrix norms induced by vector $p$-norms. We denote the set $\{1, 2, \ldots, m\}$ by $[m]$. In the defined notation system, $m$ represents the number of users or queries, $N$ denotes the total number of items, and $K$ is the number of items retrieved in the first stage. $B$ indicates the number of children nodes per non-leaf node, while $L$ specifies the number of layers in a neural network. The complete set of items is symbolized by $\mathcal{Y}$. For $i \in [m]$, $\boldsymbol{x_i} \in \mathbb{R}^d$ is the embedding vector to represent user, if we use sequence embeddings to represent the user, we denote this with a matrix $\boldsymbol{A}^{(i)}$, and $y_i \in \mathcal{Y}$ is the corresponding target item. The function $h(\boldsymbol{x}) \in \mathcal{Y}$ represents the prediction result for $\boldsymbol{x}$.

Following previous traditional notations, in the case of hierarchical classification, the hierarchy of classes $(V, E)$ is defined in the form of a rooted tree, with a root $\perp$ and a parent relationship $\pi : V\backslash\{\perp\} \to V$ where $\pi(\boldsymbol{v})$ is the parent of node $\boldsymbol{v} \in V\backslash\{\perp\}$, and $E$ denotes the set of edges with parent to child orientation. $\mathcal{B}_k(\boldsymbol{x})$ identifies nodes selected at depth $k$ during a beam search for input $\boldsymbol{x}$. For each node $\boldsymbol{v} \in V\backslash\{\perp\}$, we further define the set of its children $\mathcal{D}(\boldsymbol{v}) = \{\boldsymbol{v}' \in V\backslash\{\perp\}; \pi(\boldsymbol{v}') = \boldsymbol{v}\}$. The specialized nodes at the leaf level constitute the set of target items. Finally, for each item $y$ in $\mathcal{Y}$ we define the set of its ancestors $\mathfrak{P}(y)$ defined as

$$\mathfrak{P}(y) = \{v_1, \ldots, v_{k_y} : v_1 = \pi(y), \pi\left(v_{k_y}\right) = \perp, v_{l+1} = \pi\left(v_l\right), \forall l \in \{1, \ldots, k_y - 1\}\}.$$

## 3.2 Background

### 3.2.1 Tree Structure Retriever Model

During the retrieval stage, a tree is utilized where each leaf node represents an item. Additionally, each node in the tree has a learnable parameter vector that has the same dimensionality as the user vector. The architecture of the tree is generally determined by hierarchical clustering techniques, as shown in the studies by [29, 24].

When constructing the tree, we first need to obtain the initial item representations, which can be accomplished through various methods. The item representations can be represented by the instance-item matrix $\boldsymbol{Y} \in \{0, 1\}^{m \times N}$. A strategy to construct item representations is by leveraging indices of positive instances. For any given item $i$ within the item set $\mathcal{Y}$, its corresponding representation vector $\boldsymbol{z}_i$ is defined through normalization as: $\boldsymbol{z}_i = \overline{\boldsymbol{y}}_i / \|\overline{\boldsymbol{y}}_i\|$, where the vector $\overline{\boldsymbol{y}}_i \in \{0, 1\}^m$ signifies the $i$-th column of the instance-to-item matrix $Y$, encapsulating the relationship between instances and the item $i$. It is possible to refine the item representation by incorporating additional feature information, an enhanced formulation of $\overline{\boldsymbol{y}}_i$ is employed, expressed as $\overline{\boldsymbol{y}}_i = \sum_{j=1}^m \boldsymbol{Y}_{ji} \boldsymbol{f}_j$, where $\boldsymbol{f}_j$ denotes the feature vector associated with the $j$-th instance. Besides, some works, like [29], have employed a technique of starting with a randomly initialized tree structure aligned with item categories. The learned parameters of the leaf nodes are subsequently utilized as new initial representations for the items.

After acquiring item representations, we repeatedly apply clustering algorithms, such as $k$-means, to form the complete tree structure. In the initial phase, all items are aggregated at the root. These items are then clustered into $B$ categories, creating the root's child nodes. This procedure is recursively performed in each child node until each category is reduced to a single item, establishing the leaf nodes. A balanced distribution of items among the categories can lead to a more even tree structure, a practice widely utilized in applications.

During inference, the user representation $\boldsymbol{x}$ starts from the root node, and the path is continually extended until a leaf node is reached, using a beam search based on the model score between the user representation and the node parameter vector. Specifically, assuming a beam size of $K$, we maintain at most $K$ paths during the inference process. Initially, the user representation selects the top $K$ nodes with the highest model scores from the children of the root node, denoted as $\mathcal{B}_1(\boldsymbol{x})$. If the number of available nodes for selection is less than $K$, all available nodes are selected. $\mathcal{B}_{i+1}(\boldsymbol{x})$ denotes the selected nodes at depth $(i + 1)$ for an input $\boldsymbol{x}$,

$$\mathcal{B}_{i+1}(\boldsymbol{x}) = \mathcal{T}^K_{\boldsymbol{c} \in \mathcal{D}(\mathcal{B}_i(\boldsymbol{x}))} f(\boldsymbol{x}, \boldsymbol{c}),$$

where we denote $\mathcal{T}^K$ as the Top-K operator, which selects the top $K$ nodes from the children of nodes in $\mathcal{B}_i(\boldsymbol{x})$, denoted as $\mathcal{D}(\mathcal{B}_i(\boldsymbol{x}))$ based on the highest score $f(\boldsymbol{x}, \boldsymbol{c})$. To avoid the problem of leaf nodes lacking children during the inference process, we extend the definition of $\mathcal{D}(\boldsymbol{v})$,

$$\mathcal{D}(\boldsymbol{v}) = \{\boldsymbol{v}\}, \text{if } \boldsymbol{v} \text{ is a leaf node.}$$

In this extended definition, the children of a leaf node are considered to be the leaf node itself. The set of $K$ leaf nodes ultimately selected by this process is denoted by $\mathcal{B}(\boldsymbol{x})$.

### 3.2.2 Ranker Model

The retriever models and the ranker models are often trained independently using logged feedback data (e.g., user clicks or dwell time) generated by previous versions of the recommender system. Compared to retriever models, ranker models may utilize more contextual information to better represent the user, leading to more accurate predictions. For simplicity, in this work, we assume that both the retriever and ranker models have the same input (e.g., user interaction history sequence). The key difference is that the retriever uses a tree-structured greedy model to retrieve a subset from a large item pool, while the ranker predicts scores for each item within this subset $\mathcal{B}(\boldsymbol{x})$. Finally, the item with the highest score from $\mathcal{B}(\boldsymbol{x})$, as determined by the ranker, is returned as the prediction result, which we denote as $h(\boldsymbol{x})$.

### 3.2.3 Generalization Error

In the training of machine learning algorithms, we are constrained to a finite dataset for learning. Nevertheless, the resulting function must generalize effectively beyond the training sample. Thus, ensuring high probability guarantees for the difference between the loss on the training sample and the loss on the test population is of paramount importance. Generalization bounds aim to constrain this loss difference.

Mathematically, if we have a hypothesis class $\mathcal{H}$, sample space $\mathcal{X}$, item space $\mathcal{Y}$, loss function $\ell$, and distribution over the sample and item space $\mathcal{D}$, then our generalization gap for a set of samples and items $S = \{(\boldsymbol{x_i}, y_i)\}_{i=1}^m, \boldsymbol{x_i} \in \mathcal{X}, y_i \in \mathcal{Y}$, on the hypothesis $h \in \mathcal{H}$ is defined to be

$$\left| \mathbb{E}[\ell(h(\boldsymbol{x}), y)] - \frac{1}{m} \sum_{i=1}^m \ell(h(\boldsymbol{x_i}), y_i) \right|. \tag{1}$$

Notice how if we can have this value go to 0 with high probability over all sets of samples and for all $h \in \mathcal{H}$, then we can be confident that minimizing the empirical loss will not impact our generalization.

One tool that can be used to upper bound the generalization gap is the Rademacher complexity. The Rademacher complexity of a hypothesis class $\mathcal{H}$ is defined to be

$$\hat{\mathcal{R}}_m(\mathcal{H}) = \frac{1}{m} \mathbb{E}_{\boldsymbol{\epsilon}} \left[ \sup_{h \in \mathcal{H}} \sum_{i=1}^m \epsilon_i h(\boldsymbol{x_i}) \right],$$

where each $\epsilon_i$ are i.i.d. and take the value 1 or $-1$ each with half probability and $\boldsymbol{\epsilon} = (\epsilon_1, \ldots, \epsilon_m)$. It is well known that [21], if the magnitude of our loss function is bounded above by $c$, with probability greater than $1 - \delta$ for all $h \in \mathcal{H}$, we have

$$\left| \mathbb{E}_{(\boldsymbol{x}, y) \sim \mathcal{D}}[\ell(h(\boldsymbol{x}), y)] - \frac{1}{m} \sum_{i=1}^m \ell(h(\boldsymbol{x_i}), y_i) \right| \le 2\hat{\mathcal{R}}_m(\ell \circ \mathcal{H}) + c\sqrt{\frac{2 \log(2/\delta)}{m}}. \tag{2}$$

Therefore, if we have an upper bound on the Rademacher complexity, we can have an upper bound on the generalization gap.

# 4 Theory Results

We begin by considering the two-stage error decomposition. Let $P^{\text{err}}_{\text{two-stage}}$ denote the two-stage classification error, i.e., $P(h(\boldsymbol{x}) \neq y)$. $P^{\text{err}}_{\text{retrieve}}$ and $\tilde{P}^{\text{err}}_{\text{rank}}$ represent the classification errors caused by the retriever and ranker, respectively. We have the following proposition:

**Proposition 4.1.** *The probability of a classification error of the two-stage method can be decomposed as follows:*

$$P^{err}_{two\text{-}stage} = P^{err}_{retrieve} + \tilde{P}^{err}_{rank}(K)(1 - P^{err}_{retrieve}), \tag{3}$$

*where $P^{err}_{retrieve} = P(y \notin \mathcal{B}(\boldsymbol{x}))$ and $\tilde{P}^{err}_{rank}(K) = P(h(\boldsymbol{x}) \neq y \mid y \in \mathcal{B}(\boldsymbol{x}))$ with $|\mathcal{B}(\boldsymbol{x})| = K$.*

In Proposition 4.1, the total generalization error of two stages is decomposed into two critical components, $P^{\text{err}}_{\text{retrieve}}$ and $\tilde{P}^{\text{err}}_{\text{rank}}$. $P^{\text{err}}_{\text{retrieve}}$ captures the error when the target item isn't included in the retriever model's results, reflecting the probability that the item $y$ doesn't appear in the set $\mathcal{B}(\boldsymbol{x})$. $\tilde{P}^{\text{err}}_{\text{rank}}$ refers to the error that occurs when the target item $y$, although present in the retriever's results $\mathcal{B}(\boldsymbol{x})$, is not correctly ranked by the ranker model.

Compared to a single ranker model for classification tasks, we use $P^{\text{err}}_{\text{rank}}(N)$ to denote the classification error, where $N$ emphasizes that the ranker model is used for an $N$-class task, we have the following corollary:

**Corollary 4.2.** *The error $P^{err}_{two\text{-}stage} \leq P^{err}_{rank}(N)$ if and only if the retrieval error $P^{err}_{retrieve}$ satisfies the following inequality:*

$$P^{err}_{retrieve} \leq \frac{P^{err}_{rank}(N) - \tilde{P}^{err}_{rank}(K)}{1 - \tilde{P}^{err}_{rank}(K)}. \tag{4}$$

Regarding inequality (4), we have two approaches to enhance its validity. One is to reduce $P^{\text{err}}_{\text{retrieve}}$ without increasing the number of retriever results. The other is to improve the ranker model while keeping the retriever unchanged, to reduce $\tilde{P}^{\text{err}}_{\text{rank}}$ and thus increasing the threshold on the right side of the inequality. Furthermore, by Property 4.1, both approaches will lead to lower two-stage classification error. In the following subsections, we will analyze these two errors separately from a generalization bound perspective, revealing their relationship with the actual observed empirical errors.

## 4.1 Retriever

For the model described in Sec.3.2.1, we represent the user using the vector $\boldsymbol{x_i} \in \mathbb{R}^d$, which can be a vector representation of a text segment, or an embedding derived from a pre-trained model that includes relevant features (If we use sequence embeddings to represent the user, we denote this with a matrix $\boldsymbol{A}^{(i)}$). We consider different user and target items to be independently and identically distributed, denoted as $(\boldsymbol{x_1}, y_1), \ldots, (\boldsymbol{x_m}, y_m) \sim \mathcal{D}$.

Following previous work [1], we consider the following analogous function space induced by the function space $\mathcal{F}$:

$$\mathcal{G}_\mathcal{F} = \{g_f : (\boldsymbol{x}, y) \in \mathcal{X} \times \mathcal{Y} \mapsto \min_{\boldsymbol{v} \in \mathfrak{P}(y)} \Big( f(\boldsymbol{x}, \boldsymbol{v}) -$$
$$\max{}_{\text{K}}\{f(\boldsymbol{x}, \boldsymbol{v}') \mid \boldsymbol{v}' \in \mathcal{B}(\boldsymbol{v}, \boldsymbol{x})\} \Big) \mid f \in \mathcal{F}\}, \tag{5}$$

where $\mathfrak{P}(y)$ is the ancestors of node $y$, $\mathcal{B}(\boldsymbol{v}, \boldsymbol{x})$ denotes the set of candidates during beam search at the same level as node $\boldsymbol{v}$, in particular, if $d(\boldsymbol{v})$ represents the depth of node $\boldsymbol{v}$ within the tree structure, then $\mathcal{B}(\boldsymbol{v}, \boldsymbol{x}) = \mathcal{D}(\mathcal{B}_{d(\boldsymbol{v})-1}(\boldsymbol{x}))$, $\max_{\text{K}}$ denotes the $K$-th largest element in a set, $f(\boldsymbol{x}, \boldsymbol{v})$ represents the score function between node $\boldsymbol{v}$ and input $\boldsymbol{x}$. The specific formulation of the score function will be discussed in Section 4.1.1.

Compared with previous work, we extend the function space to the top-k form, as described in equation (5), we can observe the following proposition:

**Proposition 4.3.** *For any leaf node $y \in \mathcal{Y}$ and user representation $\boldsymbol{x}$, we have*

$$g_f(\boldsymbol{x}, y) \geq 0 \Leftrightarrow y \in \mathcal{B}(\boldsymbol{x}).$$

This implies that the probability of classification error is equal to the probability of occurrence of the event $g_f(\boldsymbol{x}, y) < 0$:

$$\mathbb{P}(y \notin \mathcal{B}(\boldsymbol{x})) = \mathbb{P}(g_f(\boldsymbol{x}, y) < 0). \tag{6}$$

For this event, we can formulate the following theorem about the Rademacher complexity of the function space $\mathcal{G}_{\mathcal{F}}$:

**Theorem 4.4.** *Consider a loss function $\mathcal{A}(x)$ that is monotonically decreasing, satisfies $\mathbb{I}(x \leq 0) \leq \mathcal{A}(x)$, and is a Lipschitz function with Lipschitz constant $c_{\mathcal{A}}$ and an upper bound $B_{\mathcal{A}}$. The following inequality holds with a probability of at least $1 - \delta$:*

$$P_{retrieve}^{err} \leq \frac{1}{m} \sum_{i=1}^{m} \mathcal{A}\left(g_f\left(\boldsymbol{x}_i, y_i\right)\right) + 4c_{\mathcal{A}}\hat{\mathcal{R}}_m(\mathcal{G}_{\mathcal{F}}) + B_{\mathcal{A}}\sqrt{\frac{2\log(2/\delta)}{m}}.$$

Theorem 4.4 presents a general result, considering an abstracted loss function under specific conditions and the Rademacher complexity of the function space. In Section 4.1.1, we will present the upper bounds of Rademacher complexity for various specific function spaces. Regarding $\mathcal{A}(g_f)$, it can be related to common loss functions, such as margin-based loss and cross-entropy. We will discuss the relationship between $\mathcal{A}(g_f)$ and these commonly used loss functions in the Appendix A.

#### 4.1.1 Effect of Model Architectures

In this part, we discuss several common score models and provide upper bounds on their Rademacher complexity.

**Linear Model**. One such model is the linear model, which is widely used in text retrieval tasks [24, 10]. It calculates scores by taking the dot product of user vectors and node vectors, which can be expressed as follows:

$$f_{\text{lin}}(\boldsymbol{x}, \boldsymbol{v}) = \langle \boldsymbol{x}, \boldsymbol{w_v} \rangle,$$

where $\boldsymbol{w_v}$ is a learnable parameter for node $v$ in the tree model. The function space $\mathcal{F}_{\text{lin}}$ is expressed as follows:

$$\mathcal{F}_{\text{lin}} = \{f : (\boldsymbol{x}, \boldsymbol{v}) \mapsto \langle \boldsymbol{x}, \boldsymbol{w_v} \rangle \mid \|\boldsymbol{w_v}\|_2 \leq B_0, \forall \, \boldsymbol{v} \in V\}. \tag{7}$$

We have the following results:

**Theorem 4.5.** *Suppose $\forall i \in [m], \|\boldsymbol{x_i}\|_2 \leq B_x$, then the Rademacher complexity of $\mathcal{G}_{\mathcal{F}_{lin}}$ can be bounded by*

$$\hat{\mathcal{R}}_m(\mathcal{G}_{\mathcal{F}_{lin}}) \leq \frac{4B_0 B_x}{\sqrt{m}}\mathcal{T},$$

*where $\mathcal{T} = BN/\sqrt{B^2 - 1}$.*

**MLP**. We consider the concatenation of the user vector and the node vector as inputs to a multilayer perceptron (MLP). This architecture is widely used in the network structures of recommender systems [8], which can be expressed as follows:

$$f_{\text{mlp}}(\boldsymbol{x}, \boldsymbol{v}) = \boldsymbol{W_L} \cdot \sigma_{L-1} \circ \sigma_{L-2} \circ \ldots \circ \sigma_1\left(\boldsymbol{x}; \boldsymbol{w_v}\right),$$

where $\boldsymbol{W_L} \in \mathbb{R}^{1 \times d_{L-1}}$, $(\boldsymbol{x}; \boldsymbol{w_v}) \in \mathbb{R}^{2d}$ represents the concatenation of the column vectors $\boldsymbol{x}$ and $\boldsymbol{w_v}$, the function $\sigma_k(\boldsymbol{x})$ is defined:

$$\sigma_k(\boldsymbol{x}) = \sigma\left(\boldsymbol{W_k}\boldsymbol{x}\right) \in \mathbb{R}^{d_k \times 1}, \; \forall k \in [L-1].$$

The function $\sigma$ is a Lipschitz continuous activation function with a Lipschitz constant $c_{\sigma}$, has the property $\sigma(0) = 0$ and $\boldsymbol{W_k} \in \mathbb{R}^{d_k \times d_{k-1}}$ represents the weight matrix. For the function space:

$$\mathcal{F}_{\text{mlp}} = \left\{f : (\boldsymbol{x}, \boldsymbol{v}) \mapsto f_{\text{mlp}}(\boldsymbol{x}, \boldsymbol{v}) \mid \|\boldsymbol{w_v}\|_2 \leq B_0, \forall \, \boldsymbol{v} \in V; \|\boldsymbol{W_k}\|_1 \leq B_1, \forall k \in [L]\right\}, \tag{8}$$

we have the following results:

**Theorem 4.6.** *Suppose $\forall i \in [m], \|\boldsymbol{x_i}\|_2 \leq B_x$, then the Rademacher complexity of $\mathcal{G}_{\mathcal{F}_{mlp}}$ can be bounded by*

$$\hat{\mathcal{R}}_m(\mathcal{G}_{\mathcal{F}_{mlp}}) \leq \frac{8c_{\sigma}^{L-1}B_1^L \cdot (B_0 + B_x)}{\sqrt{m}}\mathcal{T}.$$

**Target Attention**. As a deep neural network architecture, target attention has achieved competitive performance in recommender systems and is widely used as a score function in tree-structured recommendations[8, 29]. In contrast to the previous two models, which represent a user as a single embedding vector, the model characterizes the user representation by a history sequence of items they have interacted with. In this context, we denote the matrix of item embedding vectors that the $i$-th user has interacted with as $\boldsymbol{A}^{(i)}$,

$$\boldsymbol{A}^{(i)} = [\boldsymbol{a}_1^{(i)}, \boldsymbol{a}_2^{(i)}, ..., \boldsymbol{a}_T^{(i)}] \in \mathbb{R}^{d \times T},$$

where we consider the last $T$ recorded item interaction histories. The model uses a two-layer fully connected network to compute weights for node vectors and user-history item vectors, which can be expressed as:

$$w_k^{(i)} = \sigma \left( \boldsymbol{W}_w^{(2)} \sigma \left( \boldsymbol{W}_w^{(1)} \left[ \boldsymbol{a}_k^{(i)}; \boldsymbol{a}_k^{(i)} \odot \boldsymbol{w_v}; \boldsymbol{w_v} \right] \right) \right) \in \mathbb{R},$$

where $\boldsymbol{W}_w^{(1)} \in \mathbb{R}^{h \times 3d}$, $\boldsymbol{W}_w^{(2)} \in \mathbb{R}^{1 \times h}$, and $\sigma$ is activation function. The score function can be expressed as:

$$f_{ta}(\boldsymbol{x_i}, \boldsymbol{v}) = f_{mlp} \left( \boldsymbol{z}_1^{(i)}; \boldsymbol{z}_2^{(i)}; ...; \boldsymbol{z}_{N'}^{(i)}; \boldsymbol{w_v} \right),$$

where $\forall j \in [N']$,

$$\boldsymbol{z}_j^{(i)} = \sum_{k \in \mathcal{C}_j} w_k^{(i)} \boldsymbol{a}_k^{(i)},$$

$\mathcal{C}_j$, corresponding to different time windows, each $\mathcal{C}_j$ being mutually exclusive, satisfies the following conditions:

$$\bigcup_{j=1}^{N'} \mathcal{C}_j = \{1, 2, \ldots, T\}.$$

For the function space:

$$\mathcal{F}_{\text{ta}} = \Big\{ f : (\boldsymbol{x}, \boldsymbol{v}) \mapsto f_{\text{ta}}(\boldsymbol{x}, \boldsymbol{v}) \mid \ \|\boldsymbol{w_v}\|_2 \le B_0, \forall \, \boldsymbol{v} \in V;$$
$$\|\boldsymbol{W_k}\|_1 \le B_1, \forall k \in [L]; \ \|\boldsymbol{W}_w^{(j)}\|_1 \le B_2, \forall j \in \{1, 2\} \Big\}, \tag{9}$$

we have the following results:

**Theorem 4.7.** *Suppose $\forall i \in [m], \forall k \in [T], \|\boldsymbol{a}_k^{(i)}\|_2 \le B_a$, then the Rademacher complexity of $\mathcal{G}_{\mathcal{F}_{ta}}$ can be bounded by*

$$\hat{\mathcal{R}}_m(\mathcal{G}_{\mathcal{F}_{ta}}) \le \frac{4 c_\sigma^{L-1} B_1^L \left( B_w T + B_0 \right)}{\sqrt{m}} \mathcal{T},$$

*where $B_w = c_\sigma^2 B_2^2 \left( B_a^2 + B_a^2 B_0 + B_0 B_a \right)$.*

### 4.1.2 Insights from Generalization Bound

The theorems 4.5, 4.6, and 4.7, show the effect of three different score models on their generalization. More complex models tend to have higher function space complexity. From a generalization error perspective, this represents a tradeoff between function space complexity and empirical error, as they often result in lower empirical errors. We can see that, similar to most generalization conclusions derived from Rademacher complexity, the order of the number of sample points $m$ is $\mathcal{O}(m^{-1/2})$. This implies that as the number of samples increases, the error rate can be effectively controlled by the empirical error, resulting in a performance on the test set that is as satisfactory as on the training set.

Besides, the theoretical results reveal a relationship between model generalization capabilities with tree structure retrievers. Specifically, the generalization bound includes a term $\mathcal{O}(B/\sqrt{B^2 - 1})$, where $B$ represents the number of branches, suggesting that a tree with a larger number of child nodes (branches) tends to exhibit enhanced generalization performance. Intuitively, a tree with more branches will have a flatter structure. In an extreme case, when the number of branches equals the number of items, the tree structure becomes ineffective because it requires traversing all items during inference. This leads to the highest computational complexity, as the retriever model degenerates into

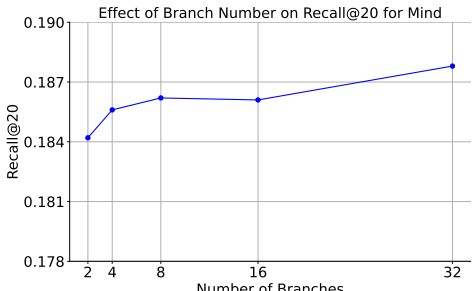
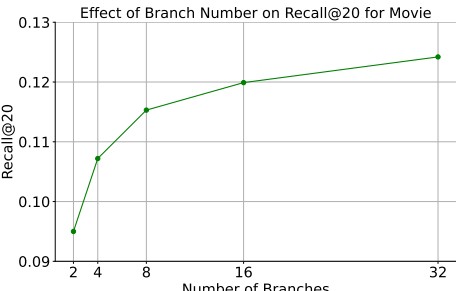

Figure 1: Effect of branch number on Recall@20 for Mind (left) and Movie (right)

a ranker model that processes the entire item pool. Thus, the number of branches of a tree structure represents, to some extent, a tradeoff between efficiency and performance.

We conduct experiments on real-world datasets to study the effects of increasing the number of tree branches. In our experiments, we use the same datasets as work [8], specifically Mind and Movie. We adopt an improved TDM model [8] as the retriever model architecture and use recall as the evaluation metric, since in the retrieval stage, the focus is on whether the target item is successfully retrieved. A more detailed description of the experimental setup can be found in Appendix D. The results are presented in Figure 1. As we can see, the recall rate increases with the number of branches. Similar phenomena can be observed in other studies related to tree models [7].

## 4.2 Ranker

In the context of training data sets $S = \{(\boldsymbol{x_1}, y_1), \ldots, (\boldsymbol{x_m}, y_m)\}$ independently and identically distributed according to distribution $\mathcal{D}$, where each $\boldsymbol{x_i} \in \mathbb{R}^d$ and $y_i \in \{1, \ldots, N\}$, we examine a subset of this data, referred to as filtered training data, given by $S' = \{(\boldsymbol{x}_1', y_1'), \ldots, (\boldsymbol{x}_{m'}', y_{m'}')\} \subset S$. We suppose this subset is independently and identically distributed, following the distribution $\mathcal{D}'$, where each $y_i' \in \mathcal{B}(\boldsymbol{x}_i')$. The generalization error of ranker $\tilde{P}_{\text{rank}}^{\text{err}}$, is defined as the expected probability of the ranking error under the distribution $\mathcal{D}'$. Specifically, we have

$$\tilde{P}_{\text{rank}}^{\text{err}} = \mathbb{E}_{(\boldsymbol{x},y)\sim\mathcal{D}'}\left[\mathbb{I}\left(f(\boldsymbol{x},y) - \max_{j\in\mathcal{B}(\boldsymbol{x})} f(\boldsymbol{x},j) < 0\right)\right],$$

where we use $f(\boldsymbol{x}, j)$ to denote the model score of user $\boldsymbol{x}$ with respect to item $j$ in this subsection.

To establish a relationship between the expected generalization error on distribution $\mathcal{D}'$ and the empirical error measured on training data distribution $\mathcal{D}$, we have the following theorem:

**Theorem 4.8.** *Consider a loss function $\Phi(\boldsymbol{x})$ that is monotonically decreasing, satisfies $\mathbb{I}(x \leq 0) \leq \Phi(\boldsymbol{x})$, and is a Lipschitz function with Lipschitz constant $c_\Phi$ and an upper bound $B_\Phi$. The following inequality holds with a probability of at least $1 - \delta$:*

$$\tilde{P}_{rank}^{err} \leq \mathbb{E}_{(\boldsymbol{x},y)\sim\mathcal{D}}\left|1 - \frac{P'(\boldsymbol{x},y)}{P(\boldsymbol{x},y)}\right| + \tilde{l}_{rank} + 4c_\Phi N (K+1) \hat{\mathcal{R}}_m(\Pi_1(\mathcal{F})) + B_\Phi \sqrt{\frac{2\log(2/\delta)}{m}},$$

*where $\Pi_1(\mathcal{F}) = \{x \mapsto f(\boldsymbol{x},y) : y \in \mathcal{Y}, f \in \mathcal{F}\}$ $P$ and $P'$ denote the probability density functions of $\mathcal{D}$ and $\mathcal{D}'$, respectively, and $\tilde{l}_{rank} = \frac{1}{m}\sum_{i=1}^m \Phi(f(\boldsymbol{x_i}, y_i) - \max_{j\in\mathcal{B}(\boldsymbol{x})} f(\boldsymbol{x_i}, j))$.*

Similar to Theorem 4.4, Theorem 4.8 presents a general result. As for the abstracted loss function $\tilde{l}_{\text{rank}}$, we can see that $\tilde{l}_{\text{rank}} \leq \hat{l}_{\text{rank}} = \frac{1}{m}\sum_{i=1}^m \Phi(f(\boldsymbol{x_i}, y_i) - \max_{j\in\mathcal{Y}} f(\boldsymbol{x_i}, j))$, where the latter, margin-based loss, is commonly used in training. This can also be extended to other common loss functions, as discussed similarly in the Appendix A. As for the Rademacher complexity of the function space, to maintain consistency with the previous subsection, we introduce the notation $\mathcal{F}^\mathcal{Y}$ to denote the restriction of the function space $\mathcal{F}$ w.r.t. $\mathcal{Y}$, specifically:

$$\mathcal{F}^\mathcal{Y} = \{f(\boldsymbol{x}, v) \in \mathcal{F} : v \in \mathcal{Y}\} \subset \mathcal{F}.$$

For the score function described in the subsection 4.1.1, we have the following theorem:

**Theorem 4.9.** *Suppose the conditions in Theorems 4.5, 4.6, and 4.7 hold, the Rademacher complexity of $\Pi_1(\mathcal{F}^{\mathcal{Y}})$ can be bounded as follows:*

$$\hat{\mathcal{R}}_m(\Pi_1(\mathcal{F}_{model}^{\mathcal{Y}})) \leq \frac{B_{model}}{\sqrt{m}},$$

*where $B_{lin} = B_0 B_x$, $B_{mlp} = 2c_\sigma^{L-1} B_1^L \cdot (B_0 + B_x)$, $B_{ta} = c_\sigma^{L-1} B_1^L (B_w T + B_0)$.*

Besides, compared to traditional generalization bounds, Theorem 4.8 includes an additional error term induced by distributional disparities. It shows that the generalization performance of the two-stage ranker model degrades due to discrepancies between the inference distribution and training distributions. When the training distribution is aligned with the inference distribution, i.e., using the subset of the training data successfully retrieved by the retriever, the distributional disparities are minimized. This suggests that in practice, aligning the training distribution and inference distribution can enhance the model's performance.

We conduct experiments on real-world datasets to verify this. In our experiments, we use the improved TDM [8] as the retriever, and the DIN model [27], which uses the target attention structure, as the ranker. We investigate the effect of the training data distribution on the ranker performance in a fixed retriever two-stage setup. The ranker model is trained in two ways: using the original training data and using a subset of training data successfully retrieved by the retriever model. A more detailed description of the experimental setup can be found in the Appendix D. We evaluate the overall classification accuracy of the two-stage model. The results are presented in Table 1. We compared the top-1 classification accuracy (i.e., Precsion@1) of rankings produced by the ranker with different numbers of retrieval items for two methods. We can see that the Harmonized Two Stage Model (H-TS) improves performance over the original Two Stage Model (TS) on these datasets.

Table 1: Comparison of classification accuracy of the two-stage model.

| Dataset | Method | K=40 | K=80 | K=120 |
|---------|--------|------|------|-------|
| Mind | TS | 0.6500 | 0.5970 | 0.5609 |
| | H-TS | **0.6565** | **0.6026** | **0.5644** |
| Movie | TS | 0.3516 | 0.3457 | 0.3453 |
| | H-TS | **0.3555** | **0.3500** | **0.3488** |

It is worth noting that while aligning the training distribution to the inference distribution eliminates the bias introduced by the distribution differences, it reduces the number of training samples available $m'$ relative to the original number of samples $m$. This reduction means that the upper bound of the generalization guarantee is also somewhat weakened. Consequently, the effectiveness of this adjustment method depends on the presence of a high recall retriever model. In our experiments, we found that a recall rate of more than $10\%$ is typically required to see an improvement effect. It must ensure that there are enough training samples to maintain the generalization performance of the model.

## 5    Conclusion

In summary, our study provides a theoretical and empirical investigation into the generalization error bounds of two-stage recommender systems, particularly emphasizing tree-based retriever models. Our study uses Rademacher complexity to analyze the generalization capabilities of several commonly used models in two-stage recommender systems, highlighting how tree models with increased branches and ranker models trained on shifted distributions can affect generalization performance. The theoretical results show that as the number of branches in the tree increases, the model tends to exhibit improved generalization capabilities, effectively balancing efficiency and accuracy. In the presence of a high recall retriever model, using a harmonized distributions to train the ranker will improve performance. Furthermore, our experimental validation on real-world datasets with advanced models for both retriever and ranker stages corroborates the theoretical insights. This study deepens the understanding of generalization in tree-based two-stage models and provides a theoretical foundation for designing more effective models in two-stage recommender systems.

## Acknowledgments and Disclosure of Funding

The work was supported by grants from the National Key R&D Program of China (No. 2021ZD0111801) and the National Natural Science Foundation of China (No. 62022077).

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

# A Effect of Loss Functions

In the discussion that follows, we examine in detail a variety of widely used loss functions, including margin-based, top-k, and soft-max losses. The critical focus of this discussion is to establish that these loss functions act either as upper bounds or as consistent multiples of the upper bounds of $\mathcal{A}(g_f)$. This perspective is to confirm the easy applicability of our theoretical results to these commonly used loss functions.

In the Theorem 4.4, for a single data point $(\boldsymbol{x}, y)$, the loss function we consider is:

$$\mathcal{A}(g_f) = \mathcal{A}\left(\min_{\boldsymbol{v} \in \mathfrak{P}(y)} \left(f(\boldsymbol{x}, \boldsymbol{v}) - \max_{\boldsymbol{v}' \in \mathcal{B}(\boldsymbol{v}, \boldsymbol{x})}{}_K f(\boldsymbol{x}, \boldsymbol{v}')\right)\right).$$

We show the following functions are upper bounds for $\mathcal{A}(g_f)$:

**Example 1** (Classical multi-class margin-based loss [5]). *The loss function is defined as*

$$\ell_{margin} = \ell\left(f(\boldsymbol{x}, \boldsymbol{v}) - \max_{\boldsymbol{v}' \neq \boldsymbol{v} \in \mathcal{B}(\boldsymbol{v}, \boldsymbol{x})} f(\boldsymbol{x}, \boldsymbol{v}')\right), \tag{10}$$

*where $\ell$ represents a function that is $c_l$-Lipschitz and monotonically decreasing. By defining $\mathcal{A}$ as $\ell$, the margin loss $\ell_{margin}$ acts as an upper bound for $\mathcal{A}(g_f)$:*

$$\mathcal{A}(g_f) \leq \max_{\boldsymbol{v} \in \mathfrak{P}(y)} \ell_{margin}.$$

**Example 2** (Top-K loss [16]). *The top-K hinge loss is defined by*

$$\ell_{topk} = \max\left\{0, \frac{1}{K}\sum_{j=1}^{K}\left(1 + f_{[j]} - f_v\right)\right\}, \tag{11}$$

*where for the sake of simplicity, we abbreviate the notation*

$$f_{[j]} := \max_{\boldsymbol{v}' \in \mathcal{B}(\boldsymbol{v}, \boldsymbol{x})}{}_j f(\boldsymbol{x}, \boldsymbol{v}'), \quad f_v := f(\boldsymbol{x}, \boldsymbol{v}). \tag{12}$$

*By setting $\mathcal{A}(\boldsymbol{x}) = \max(0, 1 - x)$, it can be observed that*

$$\mathcal{A}(g_f) = \max_{\boldsymbol{v} \in \mathfrak{P}(y)}\left(\max(0, 1 + f_{[K]} - f_v)\right) \leq \max_{\boldsymbol{v} \in \mathfrak{P}(y)} \ell_{topk}.$$

**Example 3** (Cross entropy [8]). *This loss is employed in tree-structured recommender systems and is defined as*

$$\ell_{softmax} = \sum_{\boldsymbol{v} \in \mathfrak{P}(y)} -\log\frac{\exp(f(\boldsymbol{x}, \boldsymbol{v}))}{\sum_{\boldsymbol{v}' \in \mathcal{B}(\boldsymbol{v}, \boldsymbol{x})}\exp(f(\boldsymbol{x}, \boldsymbol{v}'))}. \tag{13}$$

*Adopting the notation from equation (12), and by setting $\mathcal{A} = -\log_2 \sigma(\cdot)$, it can be observed that*

$$\mathcal{A}(g_f) = \max_{\boldsymbol{v} \in \mathfrak{P}(y)}\left\{-\log_2\left(\frac{1}{1 + \exp(f_{[K]} - f_v)}\right)\right\}$$

$$\leq \sum_{\boldsymbol{v} \in \mathfrak{P}(y)}\left\{-\log_2\left(\frac{\exp(f_v)}{\exp(f_v) + \exp(f_{[K]})}\right)\right\}$$

$$\leq \log_2(e) \cdot \ell_{softmax}.$$

# B Proof of Results

## B.1 Proof of Proposition 4.1

*Proof of Proposition 4.1.* Using the law of total probability, we have

$$P(h(\boldsymbol{x}) \neq y) = P(h(\boldsymbol{x}) \neq y \mid y \notin \mathcal{B}(\boldsymbol{x}))P(y \notin \mathcal{B}(\boldsymbol{x}))$$
$$+ P(h(\boldsymbol{x}) \neq y \mid y \in \mathcal{B}(\boldsymbol{x}))P(y \in \mathcal{B}(\boldsymbol{x}))$$
$$= P(y \notin \mathcal{B}(\boldsymbol{x})) + P(h(\boldsymbol{x}) \neq y \mid y \in \mathcal{B}(\boldsymbol{x}))\left(1 - P(y \notin \mathcal{B}(\boldsymbol{x}))\right),$$

where $P(h(\boldsymbol{x}) \neq y \mid y \notin \mathcal{B}(\boldsymbol{x}))$ is always 1. □

## B.2 Proof of Theorem 4.4

*Proof of Theorem 4.4.* Exploiting the fact that $\mathcal{A}$ dominates the $0/1$ loss and using the Rademacher data-dependent generalization bound presented in lemma C.3, we have with at least $1 - \delta$:

$$\mathbb{E}_{(\boldsymbol{x},y)\sim\mathcal{D}}\left[\mathbb{I}_{g_f(\boldsymbol{x},y)\leq 0} - 1\right] \leq \mathbb{E}_{(\boldsymbol{x},y)\sim\mathcal{D}}\left[\mathcal{A} \circ g_f(\boldsymbol{x},y) - 1\right]$$

$$\leq \frac{1}{m}\sum_{i=1}^{m}\left(\mathcal{A}\left(g_f\left(\boldsymbol{x}_i,y_i\right)\right) - 1\right) + 2\hat{\mathcal{R}}_m\left((\mathcal{A} - 1) \circ \mathcal{G}_\mathcal{F}\right) + B_\mathcal{A}\sqrt{\frac{2\log(2/\delta)}{m}},$$

where $\hat{\mathcal{R}}_m$ denotes the empirical Rademacher complexity of $(\mathcal{A} - 1) \circ \mathcal{G}_\mathcal{F}$ on $\mathcal{S}$. As $\boldsymbol{x} \mapsto \mathcal{A}(\boldsymbol{x})$ is a Lipschitz function with constant $c_\mathcal{A}$ and $(\mathcal{A} - 1)(0) = 0$. By lemma C.2 we further have:

$$\hat{\mathcal{R}}_m\left((\mathcal{A} - 1) \circ \mathcal{G}_\mathcal{F}\right) \leq 2c_\mathcal{A}\hat{\mathcal{R}}_m\left(\mathcal{G}_\mathcal{F}\right).$$

Plugging this bound into the first inequality yields the desired result:

$$\mathbb{P}\left(g_f(\boldsymbol{x},y) < 0\right) \leq \frac{1}{m}\sum_{i=1}^{m}\mathcal{A}\left(g_f\left(\boldsymbol{x}_i,y_i\right)\right) + 4c_\mathcal{A}\hat{\mathcal{R}}_m\left(\mathcal{G}_\mathcal{F},S\right) + B_\mathcal{A}\sqrt{\frac{2\log\left(2/\delta\right)}{m}}.$$

$\square$

## B.3 Proof of Theorem 4.5

The proof provided below is motivated by [1], requiring the modification for the $\max_K$ operator. In addition, we have made more detailed estimates of the results and generalize of proof technology.

**Lemma B.1.** *Define the mapping $c$ from $\mathcal{F} \times \mathcal{X} \times \mathcal{Y}$ into $V \times V$ as:*

$$c(f,\boldsymbol{x},y) = (\boldsymbol{v},\boldsymbol{v}') \Rightarrow \left(f\left(\boldsymbol{x},\boldsymbol{v}'\right) = \max_{\boldsymbol{v}''\in\mathcal{B}(\boldsymbol{v},\boldsymbol{x})} f\left(\boldsymbol{x},\boldsymbol{v}''\right)\right)$$

$$\wedge \left(f(\boldsymbol{x},\boldsymbol{v}) - f\left(\boldsymbol{x},\boldsymbol{v}'\right) = \min_{u\in\mathfrak{P}(y)}\left(f(\boldsymbol{x},u) - \max_{u'\in\mathcal{B}(u,\boldsymbol{x})} f(\boldsymbol{x},u')\right)\right),$$

*which is similar to the one given by [9] for flat multi-class classification. Then,*

$$\hat{\mathcal{R}}_m\left(\mathcal{G}_\mathcal{F}\right) \leq \frac{2}{m}\sum_{(\boldsymbol{v},\boldsymbol{v}')\in V^2}\left(\mathbb{E}_{\boldsymbol{\epsilon}}\left[\sup_{f\in\mathcal{F}}\sum_{i:c(f,\boldsymbol{x}_i,y_i)=(\boldsymbol{v},\boldsymbol{v}')}\epsilon_i\left(f\left(\boldsymbol{x}_i,\boldsymbol{v}\right)\right)\right] + \right. \tag{14}$$

$$\left.\mathbb{E}_{\boldsymbol{\epsilon}}\left[\sup_{f\in\mathcal{F}}\sum_{i:c(f,\boldsymbol{x}_i,y_i)=(\boldsymbol{v},\boldsymbol{v}')}\epsilon_i f\left(\boldsymbol{x}_i,\boldsymbol{v}'\right)\right]\right).$$

*Proof of Lemma B.1.*

$$\hat{\mathcal{R}}_m\left(\mathcal{G}_\mathcal{F}\right) = \mathbb{E}_{\boldsymbol{\epsilon}}\left[\sup_{g_f\in\mathcal{G}_\mathcal{F}}\frac{1}{m}\sum_{i=1}^{m}\epsilon_i g_f\left(\boldsymbol{x}_i,y_i\right)\right]$$

$$= \mathbb{E}_{\boldsymbol{\epsilon}}\left[\sup_{f\in\mathcal{F}}\frac{1}{m}\sum_{i=1}^{m}\epsilon_i\min_{\boldsymbol{v}\in\mathfrak{P}(y_i)}\left(f\left(\boldsymbol{x}_i,\boldsymbol{v}\right) - \max_{\boldsymbol{v}'\in\mathcal{B}(\boldsymbol{v},\boldsymbol{x})} f\left(\boldsymbol{x}_i,\boldsymbol{v}'\right)\right)\right],$$

where $\epsilon_i$ s are independent uniform random variables which take value in $\{-1, +1\}$ and are known as Rademacher variables. By construction of $c$ :

$$
\begin{aligned}
\hat{\mathcal{R}}_m\left(\mathcal{G}_{\mathcal{F}}\right) &\leq \frac{1}{m} \sum_{(\boldsymbol{v}, \boldsymbol{v}') \in V^2} \mathbb{E}_{\boldsymbol{\epsilon}}\left[\sup_{f \in \mathcal{F}} \sum_{i:c(f, \boldsymbol{x}_i, y_i)=(\boldsymbol{v}, \boldsymbol{v}')} \epsilon_i \left(f\left(\boldsymbol{x}_i, \boldsymbol{v}\right) - f\left(\boldsymbol{x}_i, \boldsymbol{v}'\right)\right)\right] \\
&\leq \frac{1}{m} \sum_{(\boldsymbol{v}, \boldsymbol{v}') \in V^2} \mathbb{E}_{\boldsymbol{\epsilon}}\left[\sup_{s \in \{-1,1\}, f \in \mathcal{F}} \sum_{i:c(f, \boldsymbol{x}_i, y_i)=(\boldsymbol{v}, \boldsymbol{v}')} s\epsilon_i f\left(\boldsymbol{x}_i, \boldsymbol{v}\right)\right. \\
&\qquad \left. + \sup_{s \in \{-1,1\}, f \in \mathcal{F}} \sum_{i:c(f, \boldsymbol{x}_i, y_i)=(\boldsymbol{v}, \boldsymbol{v}')} s\epsilon_i f\left(\boldsymbol{x}_i, \boldsymbol{v}'\right)\right] \\
&= \frac{2}{m} \sum_{(\boldsymbol{v}, \boldsymbol{v}') \in V^2} \left(\mathbb{E}_{\boldsymbol{\epsilon}}\left[\sup_{f \in \mathcal{F}} \sum_{i:c(f, \boldsymbol{x}_i, y_i)=(\boldsymbol{v}, \boldsymbol{v}')} \epsilon_i \left(f\left(\boldsymbol{x}_i, \boldsymbol{v}\right)\right)\right]\right. \\
&\qquad \left. + \mathbb{E}_{\boldsymbol{\epsilon}}\left[\sup_{f \in \mathcal{F}} \sum_{i:c(f, \boldsymbol{x}_i, y_i)=(\boldsymbol{v}, \boldsymbol{v}')} \epsilon_i f\left(\boldsymbol{x}_i, \boldsymbol{v}'\right)\right]\right),
\end{aligned}
$$

where the last equation holds because the fact $\hat{\mathcal{R}}_m(\mathcal{F} \cup -\mathcal{F}) \leq 2\hat{\mathcal{R}}_m(\mathcal{F})$. $\qquad\square$

*Proof of Theorem 4.5.* By definition, $f\left(\boldsymbol{x}_i, \boldsymbol{v}\right) - f\left(\boldsymbol{x}_i, \boldsymbol{v}'\right) = \langle \boldsymbol{w}_{\boldsymbol{v}} - \boldsymbol{w}_{\boldsymbol{v}'}, \boldsymbol{x}_i \rangle$, lemma B.1 and using Cauchy-Schwartz inequality:

$$
\begin{aligned}
\hat{\mathcal{R}}_m\left(\mathcal{G}_{\mathcal{F}}\right) &\leq \frac{2}{m} \mathbb{E}_{\boldsymbol{\epsilon}}\left[\sup_{\|\boldsymbol{W}\|_2 \leq B} \sum_{(\boldsymbol{v}, \boldsymbol{v}') \in V^2} \left|\left\langle \boldsymbol{w}_{\boldsymbol{v}} - \boldsymbol{w}_{\boldsymbol{v}'}, \sum_{i:c(f, \boldsymbol{x}_i, y_i)=(\boldsymbol{v}, \boldsymbol{v}')} \epsilon_i \boldsymbol{x}_i \right\rangle\right|\right] \\
&\leq \frac{2}{m} \mathbb{E}_{\boldsymbol{\epsilon}}\left[\sup_{\|\boldsymbol{W}\|_2 \leq B} \sum_{(\boldsymbol{v}, \boldsymbol{v}') \in V^2} \|\boldsymbol{w}_{\boldsymbol{v}} - \boldsymbol{w}_{\boldsymbol{v}'}\|_2 \left\|\sum_{i:c(f, \boldsymbol{x}_i, y_i)=(\boldsymbol{v}, \boldsymbol{v}')} \epsilon_i \boldsymbol{x}_i\right\|_2\right] \\
&\leq \frac{4B_0}{m} \sum_{(\boldsymbol{v}, \boldsymbol{v}') \in V^2} \mathbb{E}_{\boldsymbol{\epsilon}}\left[\left\|\sum_{i:c(f, \boldsymbol{x}_i, y_i)=(\boldsymbol{v}, \boldsymbol{v}')} \epsilon_i \boldsymbol{x}_i\right\|_2\right].
\end{aligned}
$$

Using Jensen's inequality, and as, $\forall i, j \in \{l \mid c\left(f, \boldsymbol{x}_i, y_i\right) = (\boldsymbol{v}, \boldsymbol{v}')\}^2, i \neq j, \mathbb{E}_{\boldsymbol{\epsilon}}\left[\epsilon_i \epsilon_j\right] = 0$, we get:

$$
\begin{aligned}
\hat{\mathcal{R}}_m\left(\mathcal{G}_{\mathcal{F}_{\text{lin}}}\right) &\leq \frac{4B_0}{m} \sum_{(\boldsymbol{v}, \boldsymbol{v}') \in V^2} \left(\mathbb{E}_{\boldsymbol{\epsilon}}\left[\left\|\sum_{i:c(f, \boldsymbol{x}_i, y_i)=(\boldsymbol{v}, \boldsymbol{v}')} \epsilon_i \boldsymbol{x}_i\right\|_2^2\right]\right)^{1/2} \\
&= \frac{4B_0}{m} \sum_{(\boldsymbol{v}, \boldsymbol{v}') \in V^2} \left(\sum_{i:c(f, \boldsymbol{x}_i, y_i)=(\boldsymbol{v}, \boldsymbol{v}')} \|\boldsymbol{x}_i\|_2^2\right)^{1/2} \\
&\leq \frac{4B_0}{m} \sum_{(\boldsymbol{v}, \boldsymbol{v}') \in V^2} \left(n_{(\boldsymbol{v}, \boldsymbol{v}')} B_x^2\right)^{1/2}.
\end{aligned}
\tag{15}
$$

We have:

$$
\sum_{(\boldsymbol{v},\boldsymbol{v}')\in V^2} \left(n_{(\boldsymbol{v},\boldsymbol{v}')}\right)^{1/2} \underset{(a)}{\leq} \sqrt{\left(\sum_{(\boldsymbol{v},\boldsymbol{v}')\in V^2} n_{(\boldsymbol{v},\boldsymbol{v}')}\right) \cdot |\{(\boldsymbol{v},\boldsymbol{v}')\in V^2 : \exists i, s.t. \, \boldsymbol{v}'\in \mathcal{B}(\boldsymbol{v},\boldsymbol{x_i})\}|}
$$
$$
\underset{(b)}{\leq} \sqrt{m}\sqrt{\sum_{\boldsymbol{v}\in V\backslash\bot}\left(\min\left(B^{d(\boldsymbol{v})-1}, N\right)-1\right)} \tag{16}
$$
$$
\underset{(c)}{\leq} \frac{\sqrt{m}BN}{\sqrt{B^2-1}},
$$

where $n_{(\boldsymbol{v},\boldsymbol{v}')} = |\{i : c\,(f,\boldsymbol{x}_i,y_i) = (\boldsymbol{v},\boldsymbol{v}')\}|$ is the number of set $\{i : c\,(f,\boldsymbol{x}_i,y_i) = (\boldsymbol{v},\boldsymbol{v}')\}$, which satisfies $\sum_{(\boldsymbol{v},\boldsymbol{v}')} n_{(\boldsymbol{v},\boldsymbol{v}')} = m$, (a) use the Cauchy-Schwartz inequality, (b) holds because for a given node $\boldsymbol{v}$, the alternative nodes $\boldsymbol{v}'$ are in the same layer of $\boldsymbol{v}$; note that this is based on a B-ary tree structure, (c) holds because

$$
\sum_{\boldsymbol{v}\in V\backslash\bot}\left(\min\left(B^{d(\boldsymbol{v})-1}, N\right)-1\right) = \sum_{d=1}^{h-2} B^{d-1}(B^{d-1}-1) + N(N-1) \leq \frac{N^2 B^2}{B^2-1},
$$

where $h$ is the depth of tree, satisfy $B^{h-2} < N \leq B^{h-1}$.

Combining equations (15) and (16), we obtain the following result:

$$
\hat{\mathcal{R}}_m\left(\mathcal{G}_{\mathcal{F}_{\mathrm{lin}}}\right) \leq \frac{4B_0 B_x}{\sqrt{m}} \frac{BN}{\sqrt{B^2-1}}.
$$

$\square$

## B.4 Proof of Theorem 4.6

Before we start the analysis, for any vectors $\boldsymbol{v},\boldsymbol{u_i}\in\mathbb{R}^d$, $\|\boldsymbol{v}\|_1 \leq B_v$, notice the following inequality:

$$
\sup_{\boldsymbol{v}} \sum_i \boldsymbol{v}^\top \boldsymbol{u_i} \leq B_v \max_{j\in[d]} \left|\sum_i \boldsymbol{e_j}\boldsymbol{u_i}\right| \leq \sum_i B_v \max_{j\in[d],s\in\{-1,1\}} s\boldsymbol{e_j}\boldsymbol{u_i}. \tag{17}
$$

*Proof of Theorem 4.6.* By lemma B.1, we have

$$
\hat{\mathcal{R}}_m\left(\mathcal{G}_{\mathcal{F}_{\mathrm{mlp}}}\right) \leq \frac{2}{m} \sum_{(\boldsymbol{v},\boldsymbol{v}')\in V^2}\left(\mathbb{E}_{\boldsymbol{\epsilon}}\left[\sup_{f\in\mathcal{F}_{\mathrm{mlp}}} \sum_{i:c(f,\boldsymbol{x}_i,y_i)=(\boldsymbol{v},\boldsymbol{v}')} \epsilon_i\left(f\left(\boldsymbol{x}_i,\boldsymbol{v}\right)\right)\right]\right.
$$
$$
\left. + \mathbb{E}_{\boldsymbol{\epsilon}}\left[\sup_{f\in\mathcal{F}_{\mathrm{mlp}}} \sum_{i:c(f,\boldsymbol{x}_i,y_i)=(\boldsymbol{v},\boldsymbol{v}')} \epsilon_i f\left(\boldsymbol{x}_i,\boldsymbol{v}'\right)\right]\right). \tag{18}
$$

Using the equation (17), we can get:

$$
\mathbb{E}_{\boldsymbol{\epsilon}}\left[\sup_{f\in\mathcal{F}_{\mathrm{mlp}}}\sum_{i:c(f,\boldsymbol{x}_i,y_i)=(\boldsymbol{v},\boldsymbol{v}')}\epsilon_i\left(f\left(\boldsymbol{x}_i,\boldsymbol{v}\right)\right)\right]
$$

$$
=\mathbb{E}_{\boldsymbol{\epsilon}}\left[\sup_{\boldsymbol{w}_{\boldsymbol{v}},\{\boldsymbol{W_k}\}_{k=1}^{L}}\sum_{i:c(f,\boldsymbol{x}_i,y_i)=(\boldsymbol{v},\boldsymbol{v}')}\epsilon_i\left\langle W_L,\sigma_{L-1}\circ\sigma_{L-2}\circ\ldots\circ\sigma_1\left(\boldsymbol{x_i};\boldsymbol{w_v}\right)\right\rangle\right]
$$

$$
\leq\|\boldsymbol{W_L}\|_1\mathbb{E}_{\boldsymbol{\epsilon}}\left[\sup_{s,j\in[d],\boldsymbol{w_v},\{\boldsymbol{W_k}\}_{k=1}^{L-1}}\sum_{i:c(f,\boldsymbol{x}_i,y_i)=(\boldsymbol{v},\boldsymbol{v}')}s\epsilon_i\left\langle\boldsymbol{e_j},\sigma_{L-1}\circ\sigma_{L-2}\circ\ldots\circ\sigma_1\left(\boldsymbol{x_i};\boldsymbol{w_v}\right)\right\rangle\right]
$$

$$
\underset{(a)}{\leq}c_{\sigma}\|\boldsymbol{W_L}\|_1\mathbb{E}_{\boldsymbol{\epsilon}}\left[\sup_{s,j\in[d],\boldsymbol{w_v},\{\boldsymbol{W_k}\}_{k=1}^{L-1}}\sum_{i:c(f,\boldsymbol{x}_i,y_i)=(\boldsymbol{v},\boldsymbol{v}')}s\epsilon_i\left\langle\boldsymbol{e_j},W_{L-1}\cdot\sigma_{L-2}\circ\ldots\circ\sigma_1\left(\boldsymbol{x_i};\boldsymbol{w_v}\right)\right\rangle\right]
$$

$$
\underset{(b)}{\leq}c_{\sigma}\|\boldsymbol{W_L}\|_1\|\boldsymbol{W_{L-1}}\|_1\mathbb{E}_{\boldsymbol{\epsilon}}\left[\sup_{s,j\in[d],\boldsymbol{w_v},\{\boldsymbol{W_k}\}_{k=1}^{L-2}}\sum_{i:c(f,\boldsymbol{x}_i,y_i)=(\boldsymbol{v},\boldsymbol{v}')}s\epsilon_i\left\langle\boldsymbol{e_j},\sigma_{L-2}\circ\ldots\circ\sigma_1\left(\boldsymbol{x_i};\boldsymbol{w_v}\right)\right\rangle\right]
$$

$$
\leq c_{\sigma}^{L-1}\Pi_{k=1}^{L}\|\boldsymbol{W_k}\|_1\mathbb{E}_{\boldsymbol{\epsilon}}\left[\sup_{s,j\in[d],\boldsymbol{w_v}}\sum_{i:c(f,\boldsymbol{x}_i,y_i)=(\boldsymbol{v},\boldsymbol{v}')}s\epsilon_i\left\langle\boldsymbol{e_j},\left(\boldsymbol{x_i};\boldsymbol{w_v}\right)\right\rangle\right]
$$

$$
\underset{(c)}{\leq}2c_{\sigma}^{L-1}\Pi_{k=1}^{L}\|\boldsymbol{W_k}\|_1\left(\underbrace{\mathbb{E}_{\boldsymbol{\epsilon}}\left[\sup_{j\in[2d]}\sum_{i:c(f,\boldsymbol{x}_i,y_i)=(\boldsymbol{v},\boldsymbol{v}')}\epsilon_i\left\langle\boldsymbol{e_j},\left(\boldsymbol{x_i};0\right)\right\rangle\right]}_{I_1}\right.
$$

$$
\left.+\underbrace{\mathbb{E}_{\boldsymbol{\epsilon}}\left[\sup_{\boldsymbol{w_v},j\in[2d]}\sum_{i:c(f,\boldsymbol{x}_i,y_i)=(\boldsymbol{v},\boldsymbol{v}')}\epsilon_i\left\langle\boldsymbol{e_j},\left(0;\boldsymbol{w_v}\right)\right\rangle\right]}_{I_2}\right),
$$

(19)

where (a) holds since $\sigma$ is applied element wise, we can bring $\boldsymbol{e}_{\boldsymbol{j}}^{\top}$ inside the function and the use of contraction inequality [11], (b) use equation (17) again as $\boldsymbol{e}_{\boldsymbol{j}}^{\top}\boldsymbol{W_{L-1}}$ is a vector, (c) holds as $\hat{\mathcal{R}}_m(\mathcal{F}\cup-\mathcal{F})\leq 2\hat{\mathcal{R}}_m(\mathcal{F})$.

As term of $I_1$, using Cauchy-Schwartz inequality and Jensen inequality:

$$
I_1\leq\left(\mathbb{E}_{\boldsymbol{\epsilon}}\left[\left\|\sum_{i:c(f,\boldsymbol{x}_i,y_i)=(\boldsymbol{v},\boldsymbol{v}')}\epsilon_i\boldsymbol{x_i}\right\|_2\right]\right)\leq\left(\mathbb{E}_{\boldsymbol{\epsilon}}\left[\left\|\sum_{i:c(f,\boldsymbol{x}_i,y_i)=(\boldsymbol{v},\boldsymbol{v}')}\epsilon_i\boldsymbol{x_i}\right\|_2^2\right]\right)^{1/2}
$$

$$
=\left(\sum_{i:c(f,\boldsymbol{x}_i,y_i)=(\boldsymbol{v},\boldsymbol{v}')}\|\boldsymbol{x_i}\|_2^2\right)^{1/2}\leq\left(n_{(\boldsymbol{v},\boldsymbol{v}')}B_x^2\right)^{1/2}.
$$

(20)

As term of $I_2$, we have:

$$I_2 \underset{(a)}{=} \mathbb{E}_{\boldsymbol{\epsilon}}\left[\sup_{\boldsymbol{w_v},j\in[2d]} \langle \boldsymbol{e_j}, (0;\boldsymbol{w_v})\rangle \cdot \left|\sum_{i:c(f,\boldsymbol{x}_i,y_i)=(\boldsymbol{v},\boldsymbol{v}')} \epsilon_i\right|\right]$$

$$\leq \mathbb{E}_{\boldsymbol{\epsilon}}\left[B_0 \cdot \left|\sum_{i:c(f,\boldsymbol{x}_i,y_i)=(\boldsymbol{v},\boldsymbol{v}')} \epsilon_i\right|\right] \tag{21}$$

$$\underset{(b)}{\leq} B_0\left(\mathbb{E}_{\boldsymbol{\epsilon}}\left[\left|\sum_{i:c(f,\boldsymbol{x}_i,y_i)=(\boldsymbol{v},\boldsymbol{v}')} \epsilon_i\right|^2\right]\right)^{1/2}$$

$$= B_0\sqrt{n_{(\boldsymbol{v},\boldsymbol{v}')}},$$

where (a) holds since for any given $(\epsilon_1,\epsilon_2,...,\epsilon_m)$, the value of $\langle \boldsymbol{e_j}, (0;\boldsymbol{w_v})\rangle$ is fixed independent of $i$, (b) use Jensen inequality.

Note that $\mathbb{E}_{\boldsymbol{\epsilon}}[\sup_{f\in\mathcal{F}_{\mathrm{mlp}}}\sum_{i:c(f,\boldsymbol{x}_i,y_i)=(\boldsymbol{v},\boldsymbol{v}')}\epsilon_i f(\boldsymbol{x}_i,\boldsymbol{v}')]$ and $\mathbb{E}_{\boldsymbol{\epsilon}}[\sup_{f\in\mathcal{F}_{\mathrm{mlp}}}\sum_{i:c(f,\boldsymbol{x}_i,y_i)=(\boldsymbol{v},\boldsymbol{v}')}\epsilon_i f(\boldsymbol{x}_i,\boldsymbol{v})]$ share the same upper bound, combined above equations (18), (19), (20), (21), and (16), we get the desired result.

$\square$

## B.5 Proof of Theorem 4.7

*Proof of Theorem 4.7.* Using the same analytical procedure as in equation (19), we can get

$$\mathbb{E}_{\boldsymbol{\epsilon}}\left[\sup_{f\in\mathcal{F}_{\mathrm{ta}}}\sum_{i:c(f,\boldsymbol{x}_i,y_i)=(\boldsymbol{v},\boldsymbol{v}')}\epsilon_i\left(f(\boldsymbol{x}_i,\boldsymbol{v})\right)\right]$$

$$= \mathbb{E}_{\boldsymbol{\epsilon}}\left[\sup_{\boldsymbol{w_v},\{\boldsymbol{W_k}\}_{k=1}^L}\sum_{i:c(f,\boldsymbol{x}_i,y_i)=(\boldsymbol{v},\boldsymbol{v}')}\epsilon_i\langle W_L, \sigma_{L-1}\circ\sigma_{L-2}\circ\ldots\circ\sigma_1(\boldsymbol{x_i};\boldsymbol{w_v})\rangle\right]$$

$$\leq c_\sigma^{L-1}\Pi_{k=1}^L\|\boldsymbol{W_k}\|_1\mathbb{E}_{\boldsymbol{\epsilon}}\left[\sup_{s,j\in[(N'+1)d],\boldsymbol{w_v},\boldsymbol{z}}\sum_{i:c(f,\boldsymbol{x}_i,y_i)=(\boldsymbol{v},\boldsymbol{v}')}s\epsilon_i\langle\boldsymbol{e_j},(\boldsymbol{z}_1;\boldsymbol{z}_2;...;\boldsymbol{z}_N;\boldsymbol{w_v})\rangle\right]$$

$$\leq c_\sigma^{L-1}\Pi_{k=1}^L\|\boldsymbol{W_k}\|_1\mathbb{E}_{\boldsymbol{\epsilon}}\left[\sum_{n=1}^{N'}\left(\sup_{s,j\in[d],\boldsymbol{z}_n}\sum_{i:c(f,\boldsymbol{x}_i,y_i)=(\boldsymbol{v},\boldsymbol{v}')}s\epsilon_i\langle\boldsymbol{e_j},\boldsymbol{z_n}\rangle\right)\right.$$

$$\left.+ \sup_{s,j\in[d],\boldsymbol{w_v}}\sum_{i:c(f,\boldsymbol{x}_i,y_i)=(\boldsymbol{v},\boldsymbol{v}')}s\epsilon_i\langle\boldsymbol{e_j},\boldsymbol{w_v}\rangle\right]. \tag{22}$$

Based on equation (21), we can obtain

$$\mathbb{E}_{\boldsymbol{\epsilon}}\left[\sup_{s,j\in[d],\boldsymbol{w_v}}\sum_{i:c(f,\boldsymbol{x}_i,y_i)=(\boldsymbol{v},\boldsymbol{v}')}s\epsilon_i\langle\boldsymbol{e_j},\boldsymbol{w_v}\rangle\right] \leq B_0\sqrt{n_{(\boldsymbol{v},\boldsymbol{v}')}}. \tag{23}$$

For other terms, we have

$$\mathbb{E}_{\boldsymbol{\epsilon}} \left[ \sup_{s,j \in [d], \boldsymbol{z_n}} \sum_{i:c(f,\boldsymbol{A}_i,y_i)=(\boldsymbol{v},\boldsymbol{v}')} s\epsilon_i \left\langle \boldsymbol{e_j}, \boldsymbol{z_n} \right\rangle \right]$$

$$= \mathbb{E}_{\boldsymbol{\epsilon}} \left[ \sup_{s,j \in [d], \boldsymbol{w}} \sum_{i:c(f,\boldsymbol{A}_i,y_i)=(\boldsymbol{v},\boldsymbol{v}')} s\epsilon_i \left\langle \boldsymbol{e_j}, \sum_{k \in \mathcal{C}_n} w_k^{(i)} \boldsymbol{a_k^{(i)}} \right\rangle \right]$$

$$\leq \sum_{k \in \mathcal{C}_n} \mathbb{E}_{\boldsymbol{\epsilon}} \left[ \sup_{s,j \in [d], \boldsymbol{w}} \sum_{i:c(f,\boldsymbol{A}_i,y_i)=(\boldsymbol{v},\boldsymbol{v}')} s\epsilon_i \left\langle \boldsymbol{e_j}, w_k^{(i)} \boldsymbol{a_k^{(i)}} \right\rangle \right]$$

$$= \sum_{k \in \mathcal{C}_n} \mathbb{E}_{\boldsymbol{\epsilon}} \left[ \sup_{s,j \in [d], \boldsymbol{w}} \sum_{i:c(f,\boldsymbol{A}_i,y_i)=(\boldsymbol{v},\boldsymbol{v}')} s\epsilon_i \sigma \left( \boldsymbol{W}_w^{(2)} \sigma \left( \boldsymbol{W}_w^{(1)} \left[ \boldsymbol{a}_k^{(i)}; \boldsymbol{a}_k^{(i)} \odot \boldsymbol{w_v}; \boldsymbol{w_v} \right] \right) \right) \left\langle \boldsymbol{e_j}, \boldsymbol{a_k^{(i)}} \right\rangle \right]$$

$$\leq c_\sigma^2 \left\| \boldsymbol{W}_w^{(1)} \right\|_1 \left\| \boldsymbol{W}_w^{(2)} \right\|_1 \sum_{k \in \mathcal{C}_n} \mathbb{E}_{\boldsymbol{\epsilon}} \left[ \sup_{s,j \in [d], j' \in [3d], \boldsymbol{w}} \right.$$

$$\left. \sum_{i:c(f,\boldsymbol{A}_i,y_i)=(\boldsymbol{v},\boldsymbol{v}')} s\epsilon_i \left\langle \boldsymbol{e_{j'}}, \left[ \boldsymbol{a}_k^{(i)}; \boldsymbol{a}_k^{(i)} \odot \boldsymbol{w_v}; \boldsymbol{w_v} \right] \right\rangle \left\langle \boldsymbol{e_j}, \boldsymbol{a_k^{(i)}} \right\rangle \right].$$
(24)

Furthermore, we can get

$$\mathbb{E}_{\boldsymbol{\epsilon}} \left[ \sup_{s,j \in [d], j' \in [3d], \boldsymbol{w}} \sum_{i:c(f,\boldsymbol{A}_i,y_i)=(\boldsymbol{v},\boldsymbol{v}')} s\epsilon_i \left\langle \boldsymbol{e_{j'}}, \left[ \boldsymbol{a}_k^{(i)}; \boldsymbol{a}_k^{(i)} \odot \boldsymbol{w_v}; \boldsymbol{w_v} \right] \right\rangle \left\langle \boldsymbol{e_j}, \boldsymbol{a_k^{(i)}} \right\rangle \right]$$

$$\leq \mathbb{E}_{\boldsymbol{\epsilon}} \left[ \sup_{s,j \in [d], j' \in [d]} \sum_{i:c(f,\boldsymbol{A}_i,y_i)=(\boldsymbol{v},\boldsymbol{v}')} s\epsilon_i \left\langle \boldsymbol{e_{j'}}, \boldsymbol{a}_k^{(i)} \right\rangle \left\langle \boldsymbol{e_j}, \boldsymbol{a_k^{(i)}} \right\rangle \right]$$

$$+ \mathbb{E}_{\boldsymbol{\epsilon}} \left[ \sup_{s,j \in [d], j' \in [d], \boldsymbol{w}} \sum_{i:c(f,\boldsymbol{A}_i,y_i)=(\boldsymbol{v},\boldsymbol{v}')} s\epsilon_i \left\langle \boldsymbol{e_{j'}}, \boldsymbol{a}_k^{(i)} \odot \boldsymbol{w_v} \right\rangle \left\langle \boldsymbol{e_j}, \boldsymbol{a_k^{(i)}} \right\rangle \right]$$
(25)

$$+ \mathbb{E}_{\boldsymbol{\epsilon}} \left[ \sup_{s,j \in [d], j' \in [d]} \sum_{i:c(f,\boldsymbol{A}_i,y_i)=(\boldsymbol{v},\boldsymbol{v}')} s\epsilon_i \left\langle \boldsymbol{e_{j'}}, \boldsymbol{w_v} \right\rangle \left\langle \boldsymbol{e_j}, \boldsymbol{a_k^{(i)}} \right\rangle \right]$$

$$= I_1 + I_2 + I_3.$$

As terms of $I_1$, we have

$$\sum_{i:c(f,\boldsymbol{A}_i,y_i)=(\boldsymbol{v},\boldsymbol{v}')} s\epsilon_i \left\langle \boldsymbol{e_{j'}}, \boldsymbol{a}_k^{(i)} \right\rangle \left\langle \boldsymbol{e_j}, \boldsymbol{a_k^{(i)}} \right\rangle = \sum_{i:c(f,\boldsymbol{A}_i,y_i)=(\boldsymbol{v},\boldsymbol{v}')} s\epsilon_i \boldsymbol{e_{j'}}^\top \boldsymbol{P}_a^{(i)} \boldsymbol{e_j}$$

$$= \sum_{i:c(f,\boldsymbol{A}_i,y_i)=(\boldsymbol{v},\boldsymbol{v}')} s\epsilon_i \operatorname{Tr}(\boldsymbol{e_j} \boldsymbol{e_{j'}}^\top \boldsymbol{P}_a^{(i)})$$

$$= \operatorname{Tr} \left( \boldsymbol{e_j} \boldsymbol{e_{j'}}^\top \left( \sum_{i:c(f,\boldsymbol{A}_i,y_i)=(\boldsymbol{v},\boldsymbol{v}')} s\epsilon_i \boldsymbol{P}_a^{(i)} \right) \right)$$

$$= \left\langle \boldsymbol{e_j} \boldsymbol{e_{j'}}^\top, \sum_{i:c(f,\boldsymbol{A}_i,y_i)=(\boldsymbol{v},\boldsymbol{v}')} s\epsilon_i \boldsymbol{P}_a^{(i)} \right\rangle_F,$$

where $\boldsymbol{P}_a^{(i)} = \boldsymbol{a}_{\boldsymbol{k}}^{(\boldsymbol{i})}\boldsymbol{a}_{\boldsymbol{k}}^{(\boldsymbol{i})\top}$. Then, we can get

$$
\begin{aligned}
I_1 &= \mathbb{E}_{\boldsymbol{\epsilon}}\left[\sup_{s,j\in[d],j'\in[d]}\ \sum_{i:c(f,\boldsymbol{A}_i,y_i)=(\boldsymbol{v},\boldsymbol{v}')} s\epsilon_i\left\langle \boldsymbol{e}_{j'},\boldsymbol{a}_k^{(i)}\right\rangle\left\langle \boldsymbol{e}_j,\boldsymbol{a}_{\boldsymbol{k}}^{(\boldsymbol{i})}\right\rangle\right] \\
&= \mathbb{E}_{\boldsymbol{\epsilon}}\left[\sup_{s,j\in[d],j'\in[d]}\ \left\langle \boldsymbol{e}_j\boldsymbol{e}_{j'}^\top,\sum_{i:c(f,\boldsymbol{A}_i,y_i)=(\boldsymbol{v},\boldsymbol{v}')} s\epsilon_i\boldsymbol{P}_a^{(i)}\right\rangle_F\right] \\
&\le \mathbb{E}_{\boldsymbol{\epsilon}}\left[\left\|\sum_{i:c(f,\boldsymbol{A}_i,y_i)=(\boldsymbol{v},\boldsymbol{v}')} \epsilon_i\boldsymbol{P}_a^{(i)}\right\|_F\right] \\
&= \sqrt{\sum_{i:c(f,\boldsymbol{A}_i,y_i)=(\boldsymbol{v},\boldsymbol{v}')} \left\|\boldsymbol{P}_a^{(i)}\right\|_F^2} \le \sqrt{n_{(\boldsymbol{v},\boldsymbol{v}')}B_a^4}.
\end{aligned}
$$

As terms of $I_2$, use the same analysis technique as for $I_1$, we can get

$$
\begin{aligned}
I_2 &= \mathbb{E}_{\boldsymbol{\epsilon}}\left[\sup_{s,j\in[d],j'\in[d],\boldsymbol{w}}\ \sum_{i:c(f,\boldsymbol{A}_i,y_i)=(\boldsymbol{v},\boldsymbol{v}')} s\epsilon_i\left\langle \boldsymbol{e}_{j'},\boldsymbol{a}_k^{(i)}\odot\boldsymbol{w}_{\boldsymbol{v}}\right\rangle\left\langle \boldsymbol{e}_j,\boldsymbol{a}_{\boldsymbol{k}}^{(\boldsymbol{i})}\right\rangle\right] \\
&= \mathbb{E}_{\boldsymbol{\epsilon}}\left[\sup_{s,j\in[d],j'\in[d],\boldsymbol{w}}\ \sum_{i:c(f,\boldsymbol{A}_i,y_i)=(\boldsymbol{v},\boldsymbol{v}')} s\epsilon_i\left\langle \boldsymbol{e}_{j'}\odot\boldsymbol{w}_{\boldsymbol{v}},\boldsymbol{a}_k^{(i)}\right\rangle\left\langle \boldsymbol{e}_j,\boldsymbol{a}_{\boldsymbol{k}}^{(\boldsymbol{i})}\right\rangle\right] \\
&= \mathbb{E}_{\boldsymbol{\epsilon}}\left[\sup_{s,j\in[d],j'\in[d],\boldsymbol{w}}\ \left\langle \boldsymbol{e}_j\boldsymbol{e}_{j'}^\top\odot\boldsymbol{w}_{\boldsymbol{v}}^\top,\sum_{i:c(f,\boldsymbol{A}_i,y_i)=(\boldsymbol{v},\boldsymbol{v}')} s\epsilon_i\boldsymbol{P}_a^{(i)}\right\rangle_F\right] \\
&\le \sqrt{n_{(\boldsymbol{v},\boldsymbol{v}')}B_a^4B_0^2}.
\end{aligned}
$$

As terms of $I_3$, we have

$$
\begin{aligned}
I_3 &= \mathbb{E}_{\boldsymbol{\epsilon}}\left[\sup_{s,j\in[d],j'\in[d],\boldsymbol{w}}\ \sum_{i:c(f,\boldsymbol{A}_i,y_i)=(\boldsymbol{v},\boldsymbol{v}')} s\epsilon_i\left\langle \boldsymbol{e}_{j'},\boldsymbol{w}_{\boldsymbol{v}}\right\rangle\left\langle \boldsymbol{e}_j,\boldsymbol{a}_{\boldsymbol{k}}^{(\boldsymbol{i})}\right\rangle\right] \\
&= \mathbb{E}_{\boldsymbol{\epsilon}}\left[\sup_{s,j\in[d],j'\in[d],\boldsymbol{w}}\ \left\langle \boldsymbol{e}_{j'},\boldsymbol{w}_{\boldsymbol{v}}\right\rangle\left\langle \boldsymbol{e}_j,\sum_{i:c(f,\boldsymbol{A}_i,y_i)=(\boldsymbol{v},\boldsymbol{v}')} s\epsilon_i\boldsymbol{a}_k^{(i)}\right\rangle\right] \\
&\le B_0\mathbb{E}_{\boldsymbol{\epsilon}}\left[\left\|\sum_{i:c(f,\boldsymbol{A}_i,y_i)=(\boldsymbol{v},\boldsymbol{v}')} \epsilon_i\boldsymbol{a}_{\boldsymbol{k}}^{(\boldsymbol{i})}\right\|_2\right] \le B_0\sqrt{n_{(\boldsymbol{v},\boldsymbol{v}')}B_a^2}.
\end{aligned}
$$

Combined equations (22), (23), (24), (25), and use $\sum_{n=1}^{N'}|\mathcal{C}_n| = T$ we have

$$
\begin{aligned}
&\mathbb{E}_{\boldsymbol{\epsilon}}\left[\sup_{f\in\mathcal{F}_{\text{ta}}}\ \sum_{i:c(f,\boldsymbol{x}_i,y_i)=(\boldsymbol{v},\boldsymbol{v}')} \epsilon_i\left(f\left(\boldsymbol{x}_i,\boldsymbol{v}\right)\right)\right] \\
&\le c_\sigma^{L-1}\Pi_{k=1}^L\|\boldsymbol{W}_{\boldsymbol{k}}\|_1\left(c_\sigma^2\left\|\boldsymbol{W}_w^{(1)}\right\|_1\left\|\boldsymbol{W}_w^{(2)}\right\|_1\left(B_a^2+B_a^2B_0+B_0B_a\right)T+B_0\right)\sqrt{n_{(\boldsymbol{v},\boldsymbol{v}')}}.
\end{aligned}
\tag{26}
$$

Note that

$$
\mathbb{E}_{\boldsymbol{\epsilon}}\left[\sup_{f\in\mathcal{F}_{\text{ta}}}\ \sum_{i:c(f,\boldsymbol{x}_i,y_i)=(\boldsymbol{v},\boldsymbol{v}')} \epsilon_if\left(\boldsymbol{x}_i,\boldsymbol{v}'\right)\right]
$$

and

$$\mathbb{E}_{\boldsymbol{\epsilon}} \left[ \sup_{f \in \mathcal{F}_{\mathrm{ta}}} \sum_{i:c(f, \boldsymbol{x}_i, y_i)=(\boldsymbol{v}, \boldsymbol{v}')} \epsilon_i f\left(\boldsymbol{x}_i, \boldsymbol{v}\right) \right]$$

share the same upper bound, combined lemma B.1, equations (26), and (16), we get the desired result.

$\square$

## B.6 Proof of Theorem 4.8

$$\tilde{P}_{\mathrm{rank}}^{\mathrm{err}}(K) = \mathbb{E}_{(\boldsymbol{x}, y) \sim \mathcal{D}'} \left[ \mathbb{I} \left[ f(\boldsymbol{x_i}, y_i) - \max_{j \in \mathcal{B}(\boldsymbol{x})} f(\boldsymbol{x}, j) < 0 \right] \right]$$

$$= \mathbb{E}_{(\boldsymbol{x}, y) \sim \mathcal{D}} \left[ \mathbb{I} \left[ f(\boldsymbol{x_i}, y_i) - \max_{j \in \mathcal{B}(\boldsymbol{x})} f(\boldsymbol{x}, j) < 0 \right] \cdot \frac{P'(\boldsymbol{x}, y)}{P(\boldsymbol{x}, y)} \right] \qquad (27)$$

$$\leq \mathbb{E}_{(\boldsymbol{x}, y) \sim \mathcal{D}} \left| 1 - \frac{P'(\boldsymbol{x}, y)}{P(\boldsymbol{x}, y)} \right| + \mathbb{E}_{(\boldsymbol{x}, y) \sim \mathcal{D}} \left[ 1_{\rho_f(\boldsymbol{x}, y) < 0} \right],$$

where we define

$$\rho_f(\boldsymbol{x}, y) = \min_{y' \in \mathcal{B}(\boldsymbol{x})} \left( f(\boldsymbol{x}, y) - f\left(\boldsymbol{x}, y'\right) \right).$$

Let $\widetilde{\mathcal{F}} = \{(\boldsymbol{x}, y) \mapsto \rho_f(\boldsymbol{x}, y) : f \in \mathcal{F}\}$, By lemma C.3, with probability at least $1 - \delta$, for all $f \in \mathcal{F}$:

$$\mathbb{E}\left[\Phi\left(\rho_f(\boldsymbol{x}, y)\right)\right] \leq \frac{1}{m} \sum_{i=1}^{m} \Phi\left(\rho_f\left(\boldsymbol{x_i}, y_i\right)\right) + 2\hat{\mathcal{R}}_m(\Phi \circ \tilde{\mathcal{F}}) + B_\Phi \sqrt{\frac{2 \log\left(2/\delta\right)}{m}}.$$

Since $\mathbb{I}(u < 0) \leq \Phi(u)$ for all $u \in \mathbb{R}$, and given the Lipschitz continuity of $\Phi$, we can write:

$$\mathbb{E}\left[1_{\rho_f(\boldsymbol{x}, y) < 0}\right] \leq \mathbb{E}\left[\Phi\left(\rho_f(\boldsymbol{x}, y)\right)\right] \leq \frac{1}{m} \sum_{i=1}^{m} \Phi\left(\rho_f\left(\boldsymbol{x_i}, y_i\right)\right) + 4c_\Phi \hat{\mathcal{R}}_m(\widetilde{\mathcal{F}}) + B_\Phi \sqrt{\frac{2 \log\left(2/\delta\right)}{m}}.$$

$$(28)$$

$\hat{\mathcal{R}}_m(\widetilde{\mathcal{F}})$ can be upper-bounded as follows:

$$\hat{\mathcal{R}}_m(\widetilde{\mathcal{F}}) = \frac{1}{m} \mathbb{E}_{\boldsymbol{\epsilon}} \left[ \sup_{f \in \mathcal{F}} \sum_{i=1}^{m} \epsilon_i \left( f\left(\boldsymbol{x_i}, y_i\right) - \max_{y \in \mathcal{B}(\boldsymbol{x_i})} f\left(\boldsymbol{x_i}, y\right) \right) \right]$$

$$\leq \frac{1}{m} \mathbb{E}_{\boldsymbol{\epsilon}} \left[ \sup_{f \in \mathcal{F}} \sum_{i=1}^{m} \epsilon_i f\left(\boldsymbol{x_i}, y_i\right) \right] + \frac{1}{m} \mathbb{E}_{\boldsymbol{\epsilon}} \left[ \sup_{f \in \mathcal{F}} \sum_{i=1}^{m} \epsilon_i \max_{y \in \mathcal{B}(\boldsymbol{x_i})} \left( f\left(\boldsymbol{x_i}, y\right) \right) \right].$$

Now we bound the first term above. Observe that

$$\frac{1}{m} \mathbb{E}_{\boldsymbol{\epsilon}} \left[ \sup_{f \in \mathcal{F}} \sum_{i=1}^{m} \epsilon_i f\left(\boldsymbol{x_i}, y_i\right) \right] = \frac{1}{m} \mathbb{E}_{\boldsymbol{\epsilon}} \left[ \sup_{f \in \mathcal{F}} \sum_{i=1}^{m} \sum_{y \in \mathcal{Y}} \epsilon_i f\left(\boldsymbol{x_i}, y\right) 1_{y_i = y} \right]$$

$$\leq \frac{1}{m} \sum_{y \in \mathcal{Y}} \mathbb{E}_{\boldsymbol{\epsilon}} \left[ \sup_{f \in \mathcal{F}} \sum_{i=1}^{m} \epsilon_i f\left(\boldsymbol{x_i}, y\right) 1_{y_i = y} \right] \qquad (29)$$

$$= \sum_{y \in \mathcal{Y}} \frac{1}{m} \mathbb{E}_{\boldsymbol{\epsilon}} \left[ \sup_{f \in \mathcal{F}} \sum_{i=1}^{m} \epsilon_i f\left(\boldsymbol{x_i}, y\right) \left( \frac{s_i}{2} + \frac{1}{2} \right) \right],$$

where $s_i = 2 \cdot 1_{y_i = y} - 1$. Since $\epsilon_i \in \{-1, +1\}$, we have that $\epsilon_i$ and $\epsilon_i s_i$ admit the same distribution and, for any $y \in \mathcal{Y}$, each of the terms of the right-hand side can be bounded as follows:

$$\frac{1}{m} \mathbb{E}_{\boldsymbol{\epsilon}} \left[ \sup_{f \in \mathcal{F}} \sum_{i=1}^{m} \epsilon_i f\left(\boldsymbol{x_i}, y\right) \left( \frac{s_i}{2} + \frac{1}{2} \right) \right]$$

$$\leq \frac{1}{2m} \mathbb{E}_{\boldsymbol{\epsilon}} \left[ \sup_{f \in \mathcal{F}} \sum_{i=1}^{m} \epsilon_i s_i f\left(\boldsymbol{x_i}, y\right) \right] + \frac{1}{2m} \mathbb{E}_{\boldsymbol{\epsilon}} \left[ \sup_{f \in \mathcal{F}} \sum_{i=1}^{m} \epsilon_i f\left(\boldsymbol{x_i}, y\right) \right] \qquad (30)$$

$$\leq \hat{\mathfrak{R}}_m\left(\Pi_1(\mathcal{F})\right).$$

Thus, we can write $\frac{1}{m}\mathbb{E}_{\boldsymbol{\epsilon}}\left[\sup_{f\in\mathcal{F}}\sum_{i=1}^{m}\epsilon_i f\left(\boldsymbol{x_i}, y_i\right)\right] \leq N\hat{\mathcal{R}}_m\left(\Pi_1(\mathcal{F})\right)$. To bound the second term, we first apply lemma C.5 which yields that

$$\frac{1}{m}\mathbb{E}_{\boldsymbol{\epsilon}}\left[\sup_{f\in\mathcal{F}}\sum_{i=1}^{m}\epsilon_i \max_{y\in\mathcal{B}(\boldsymbol{x_i})} f\left(\boldsymbol{x_i}, y\right)\right] \leq \sum_{j=1}^{K}\hat{\mathcal{R}}_m\left(\mathcal{F}_j\right),$$

where we use $\mathcal{B}_j(\boldsymbol{x_i})$ denote the $j$-th elements in $\mathcal{B}(\boldsymbol{x_i})$ and

$$\begin{aligned}
\hat{\mathcal{R}}_m\left(\mathcal{F}_j\right) &= \frac{1}{m}\mathbb{E}_{\boldsymbol{\epsilon}}\left[\sup_{f\in\mathcal{F}}\sum_{i=1}^{m}\epsilon_i f\left(\boldsymbol{x_i}, \mathcal{B}_j(\boldsymbol{x_i})\right)\right] \\
&\leq \frac{1}{m}\sum_{y\in\mathcal{Y}}\mathbb{E}_{\boldsymbol{\epsilon}}\left[\sup_{f\in\mathcal{F}}\sum_{i=1}^{m}\epsilon_i f\left(\boldsymbol{x_i}, y\right)\mathbb{I}(\mathcal{B}_j(\boldsymbol{x_i}) = y)\right] \\
&\leq N\hat{\mathcal{R}}_m\left(\Pi_1(\mathcal{F})\right),
\end{aligned}$$

where the last inequality holds due to equations (29) and (30).

Combined equations (27), (28) and above equations, we get the desired results.

### B.7 Proof of Theorem 4.9

Using the same analysis applied to equation 15, we can derive the following results:

$$\mathbb{E}_{\boldsymbol{\epsilon}}\left[\sup_{f\in\mathcal{F}_{\text{lin}}, v\in\mathcal{Y}}\sum_{i=1}^{m}\epsilon_i\left(f\left(\boldsymbol{x_i}, \boldsymbol{v}\right)\right)\right] \leq B_0 B_x \sqrt{m}.$$

Using the same analysis applied to equation 19, we can derive the following results:

$$\mathbb{E}_{\boldsymbol{\epsilon}}\left[\sup_{f\in\mathcal{F}_{\text{mlp}}, v\in\mathcal{Y}}\sum_{i=1}^{m}\epsilon_i\left(f\left(\boldsymbol{x_i}, \boldsymbol{v}\right)\right)\right] \leq 2c_\sigma^{L-1}B_1^L\cdot(B_0 + B_x)\sqrt{m}.$$

Using the same analysis applied to equation 26, we can derive the following results:

$$\begin{aligned}
&\mathbb{E}_{\boldsymbol{\epsilon}}\left[\sup_{f\in\mathcal{F}_{\text{ta}}, v\in\mathcal{Y}}\sum_{i=1}^{m}\epsilon_i\left(f\left(\boldsymbol{x_i}, \boldsymbol{v}\right)\right)\right] \\
&\leq c_\sigma^{L-1}\Pi_{k=1}^{L}\|\boldsymbol{W_k}\|_1\left(c_\sigma^2\left\|\boldsymbol{W}_w^{(1)}\right\|_1\left\|\boldsymbol{W}_w^{(2)}\right\|_1\left(B_a^2 + B_a^2 B_0 + B_0 B_a\right)T + B_0\right)\sqrt{m}.
\end{aligned}$$

## C Auxiliary Lemmas

**Lemma C.1.** *For matrix $\boldsymbol{A}$ and $\boldsymbol{B}$, vector $\boldsymbol{v}$, we have*

$$\|\boldsymbol{ABv}\|_\infty \leq \|\boldsymbol{A}\|_\infty\|\boldsymbol{B}\|_\infty\|\boldsymbol{v}\|_\infty,$$

*where we denoted $\|\boldsymbol{A}\|_\infty$ as $\|\boldsymbol{A}\|_{\infty,\infty} = max(|a_{i,j}|)$.*

*Proof of lemma C.1.* We have

$$\|\boldsymbol{ABv}\|_\infty = \left\|\boldsymbol{A}\frac{\boldsymbol{Bv}}{\|\boldsymbol{Bv}\|_\infty}\right\|_\infty\|\boldsymbol{Bv}\|_\infty \leq \|\boldsymbol{A}\|_\infty\|\boldsymbol{Bv}\|_\infty \leq \|\boldsymbol{A}\|_\infty\|\boldsymbol{B}\|_\infty\|\boldsymbol{v}\|_\infty.$$

$\square$

**Lemma C.2** (Theorem 4.15 (iv) in [22]). *Let $\mathcal{F}$ be classes of real functions. If $\mathcal{A} : \mathbb{R} \to \mathbb{R}$ is Lipschitz with constant $L$ and satisfies $\mathcal{A}(0) = 0$, then*

$$\hat{R}_\ell(\mathcal{A} \circ \mathcal{F}) \leq 2L\hat{R}_\ell(\mathcal{F}).$$

**Lemma C.3** (Theorem 26.5 in [21]). *If the magnitude of our loss function is bounded above by $c$, with probability greater than $1 - \delta$ for all $h \in \mathcal{H}$, we have*

$$\left| \mathbb{E}_{(\boldsymbol{x},y)\sim\mathcal{D}}[\ell(h(\boldsymbol{x}),y)] - \frac{1}{n}\sum_{i=1}^{n}\ell\left(h\left(\boldsymbol{x_i}\right),y_i\right) \right| \leq 2\hat{\mathcal{R}}_n(\ell \circ \mathcal{H}, S) + c\sqrt{\frac{2\log(2/\delta)}{n}},$$

*where $\ell \circ \mathcal{H} = \{\ell(h(\boldsymbol{x}),y) \mid (\boldsymbol{x},y) \in \mathcal{X} \times \mathcal{Y}, h \in \mathcal{H}\}$.*

**Lemma C.4** (Contraction lemma, lemma 26.9 in [21]). *For each $i \in [m]$, let $\phi_i : \mathbb{R} \to \mathbb{R}$ be a $\rho$ Lipschitz function, namely for all $\alpha, \beta \in \mathbb{R}$ we have $|\phi_i(\alpha) - \phi_i(\beta)| \leq \rho|\alpha - \beta|$. For $\boldsymbol{a} \in \mathbb{R}^m$ let $\boldsymbol{\phi}(\boldsymbol{a})$ denote the vector $(\phi_1\left(a_1\right), \ldots, \phi_m\left(y_m\right))$. Let $\boldsymbol{\phi} \circ A = \{\boldsymbol{\phi}(\boldsymbol{a}) : a \in A\}$. Then,*

$$\hat{\mathcal{R}}_m(\phi \circ A) \leq \rho\hat{\mathcal{R}}_m(A).$$

**Lemma C.5** ( Lemma 9.1 in [20]). *Let $\mathcal{F}_1, \ldots, \mathcal{F}_l$ be $l$ hypothesis sets, $l \geq 1$, and let $\mathcal{G} = \{\max\{h_1, \ldots, h_l\} : h_i \in \mathcal{F}_i, i \in [l]\}$. Then, for any sample $S$ of size $m$, the empirical Rademacher complexity of $\mathcal{G}$ can be upper bounded as follows:*

$$\widehat{\mathfrak{R}}_m(\mathcal{G}) \leq \sum_{j=1}^{l} \widehat{\mathfrak{R}}_m\left(\mathcal{F}_j\right).$$

# D    Experiments

We conduct experiments on real-world datasets to validate the effectiveness of the proposed method and theoretical insights. All experiments were conducted on a Linux server equipped with a 3.00 GHz Intel CPU, 300 GB of main memory, and NVIDIA 20/30 series GPUs.

Table 2: Statistics of Datasets

| Dataset | #User | #Item | #Interaction | Density |
|---------|-------|-------|--------------|---------|
| Mind | 36,281 | 7,129 | $5,610,960$ | $2.16\%$ |
| Movie | 69,878 | 10,677 | $10,000,054$ | $1.34\%$ |

## D.1    Datasets

We evaluate the two-stage models with two real-world recommendation datasets, which can be downloaded from the url[2]. The datasets are MovieLens 10M (abbreviated as Movie), Microsoft News Dataset (abbreviated as Mind). Following previous work [8, 7], since some datasets only include rating-based explicit feedback, they should be converted into implicit feedback for inputs. These datasets are pre-processed by filtering the users who interact with no more than 15 items. The overall information of datasets is summarized in Table 2.

## D.2    Settings

In each dataset, we randomly choose $10\%$ users as validation users, $10\%$ users as test users, and all the left users as training users. Following [29, 28], we use a slide window to split user-item interaction histories into slices of length 70 at most. For training users' data, the first 69 interactions are used for input context and the 70-th item is regarded as the ground truth of prediction. Each of the 10 time windows contains $[1, 1, 1, 2, 2, 2, 10, 10, 20, 20]$ (sum up to 69; if the length of behavior history is less than 69, pad the absence by zero) interactions. For data of both validation users and test users, we regard the first half as context and others as ground truth.

In the experiment on the effect of branch number, reported in Figure 1, we used a structure similar to the TDM model [29], except that we increased the number of tree branches and replaced the loss used in training with sampled softmax like [8]. We randomly initialize the correspondence between leaf nodes and labels and use a single well-trained tree model as the retriever model. The dimensions of item and node embeddings are both set to 24 across different branch numbers. We use Adam as the optimizer, with a learning rate of $1.0e\text{-}3$ with exponential decay. For different branch numbers,

---

[2]`https://drive.google.com/drive/folders/1ahiLmzU7cGRPXf5qGMqtAChte2eYp9gI`

we conduct a grid search on the hyperparameters, including weight decay and the number of negative samples. Specifically, we explore weight decay values within the range $[1e\text{-}2, 1e\text{-}3, 1e\text{-}4, 1e\text{-}5]$, and the number of negative samples ranges from 50 to 200 in increments of 10. For each branch number, we perform a beam search with a beam size of 100 and report the highest $recall@20$ value achieved during the grid search of hyperparameters.

In the experiment on harmonized distribution reported in Table 1, we analyze the effect of the training data distribution on the performance of the ranker model within the two-stage model. Following the same setup as the previous experiment, we use the same retriever model and the DIN model [27] as the ranker model. In DIN, we replace the original loss function with a sampled softmax loss, sampling 60 negative examples for each loss computation. The embedding dimension of each item is set to 96, the hidden dimensions of the attention units are set to $[64, 16]$, and the hidden dimensions of the fully connected layers are set to $[200, 80, 1]$. The retriever model remains fixed throughout the experiment, while the ranker model is trained on different datasets: the original training data and a subset containing only items successfully retrieved by the retriever. We use precision as a metric to save the ranker model that performs best on the validation set during the training process and then evaluate the overall classification accuracy on the test set of the two-stage model.

