# OpenReview forum: "Generalization Error Bounds for Two-stage Recommender Systems with Tree Structure"
_NeurIPS.cc/2024/Conference — NeurIPS 2024 oral_

### Official Review · Reviewer_g6F7 · 2024-07-08

**Soundness:** 3
**Presentation:** 4
**Contribution:** 3
**Rating:** 7
**Confidence:** 5

**Summary:**

This paper presents the first generalization analysis of the learning algorithm driving the two-stage recommender systems. Specifically, it considers a representative two-stage recommender system with a tree structure, which consists of an efficient tree-based retriever and a more precise yet time-consuming ranker. An error decomposition framework is proposed, based on which the Rademacher complexity is applied to derive generalization upper bounds for various tree-based retrievers using beam search, as well as for different ranker models under a shifted training distribution. The upper bounds indicate that increasing the branches in tree-based retrievers and harmonizing distributions across stages can enhance the generalization performance of two-stage recommender systems. Furthermore, this theoretical finding is validated by experiments on real-world datasets.

**Strengths:**

This paper studies a timely topic. The two-stage recommender systems are widely adopted in industry and achieve remarkable empirical performance in balancing the computational efficiency and recommendation accuracy yet lack theoretical understanding on why it works so well. It fills in this gap and provides a nice attempt for this topic.

The derived generalization upper bounds are meaningful, revealing valuable insights. The derived generalization upper bounds indicate that optimizing the design choice of two-stage recommender systems. For example, increasing the branches in tree-based retrievers and harmonizing distributions across stages can enhance the generalization performance of two-stage recommender systems.

Theoretical findings are aligned with empirical experiments on real-world datasets. This further demonstrates that the generalization upper bounds are useful and powerful.

The generalization upper bounds are technically nontrivial and the proof idea looks general and systematic. They nicely characterize the impact of various design choices of two-stage recommender systems on the generalization performance. They have a high potential to inspire the generalization analysis of other variants of two-stage recommender systems.

**Weaknesses:**

This paper lacks a discussion on the potential limitations of the derived generalization upper bounds. I believe that this would make a good complement to the contribution.

**Questions:**

Can the authors comment on the potential limitations of the derived generalization upper bounds?

**Limitations:**

The authors use the Rademacher complexity as a tool for proving the generalization upper bounds. The potential limitations of this work lie in the limitations of the Rademacher complexity tool itself. I do not think it has a negative impact on the contribution of this work.

---

> ### Author Rebuttal · Authors · 2024-08-05
>
> We greatly appreciate your high recognition of our work, and are eager to share our thoughts with you.
>
> **Response to Questions 1:**
>
> > **Questions 1:** Can the authors comment on the potential limitations of the derived generalization upper bounds?
>
> Thank you for your question. While the derived generalization bounds offer valuable insights, they come with some limitations. The tradeoff between model complexity and computational efficiency is a key consideration, as more complex models can lead to higher computational costs despite improved generalization. In addition, aligning the training distribution with the inference distribution reduces distributional bias, but also reduces the number of training samples, which may weaken the generalization guarantee. Thus, in practical applications, there is a need for careful management of these tradeoffs.

---

> > ### Comment · Reviewer_g6F7 · 2024-08-14
> > **Thank you for the authors' responses**
> >
> > I appreciate the authors' response. My concerns have been addressed satisfactorily, and I maintain my positive evaluation of this paper.

---

### Official Review · Reviewer_homr · 2024-07-12

**Soundness:** 3
**Presentation:** 2
**Contribution:** 2
**Rating:** 7
**Confidence:** 3

**Summary:**

This paper studies two-stage recommender systems. The authors focus on analyzing the generalization error of the retriever and ranker components within a two-stage recommendation model, specifically examining the Rademacher complexity. The findings are supported by both theoretical analysis and empirical studies

**Strengths:**

1. The paper addresses a common structure used in recommender systems, i.e., the two-stage model. It is also used in other machine learning tasks, demonstrating broader impact.
2. The analysis of generalization errors in two-stage models fills a gap in the existing literature, which has predominantly focused on efficiency-related issues such as convergence rates. This paper enhances the understanding of this model.
3. The authors investigate the impact of different scoring models on the tree retriever model. They demonstrate that different model structures result in varying levels of generalization errors. This experimental validation inspired by the theoretical analysis further supports the findings.

**Weaknesses:**

1. Although this paper is generally well-written, I suggest the authors create a separate "Experiments" section and include a list of notations to enhance clarity.
2. Please provide more detailed explanations of "Harmonized distribution" and "Harmonized model".

**Questions:**

None

**Limitations:**

Although limitations are mentioned in theoretical result discussion, the authors are encouraged to create a separate "Limitations" section.

---

> ### Author Rebuttal · Authors · 2024-08-05
>
> Thank you for your recognition of our work and providing the valuable suggestions and constructive comments.
>
> **Response to Weakness 1:**
>
> > **Weakness 1:** Although this paper is generally well-written, I suggest the authors create a separate "Experiments" section and include a list of notations to enhance clarity.
>
> Thanks for your suggestion. Due to space limitations, we initially decided to organize the paper as it is. In future revisions, we will consider creating a separate "Experiments" section to better organize the content. Additionally, we will include a list of notations to enhance readability and understanding of the paper.
>
> **Response to Weakness 2:**
>
> > **Weakness 2:** Please provide more detailed explanations of "Harmonized distribution" and "Harmonized model".
>
> Thanks for your comment. In our work, the "Harmonized distribution" arises within the context of the ranker model. During model inference, the ranker model deals with a distribution of data that has been filtered by the retriever. We name the distribution of successfully retrieved data the "Harmonized distribution," denoted as $\mathcal{D^\prime}$ in the paper. This distribution is harmonized with the retriever model, and its importance lies in the fact that the accuracy of the ranker in this stage is directly influenced by this data distribution.
>
> The term "Harmonized model" in our work refers to a two-stage model where the ranker model is trained on the Harmonized distribution. Unlike typical two-stage models that are trained independently, the Harmonized model eliminates the additional bias caused by different target data distributions, leading to better overall performance.

---

### Official Review · Reviewer_1EMB · 2024-07-12

**Soundness:** 3
**Presentation:** 3
**Contribution:** 3
**Rating:** 7
**Confidence:** 4

**Summary:**

This paper theoretically analyzes the generalization bounds of two-stage recommender systems using Rademacher complexity. It examines the generalization bounds of the tree-structured retriever and the subsequent ranker, respectively. The paper concludes that the more branches a tree-structured retriever has, the tighter the corresponding generalization bound. Additionally, the smaller the discrepancy between the training distribution and the inference distribution of the ranker, the tighter the generalization bound. The authors also conducted experiments to validate their theoretical findings. Overall, this is a solid work, providing convincing theoretical evidence and offering insightful suggestions.

**Strengths:**

1. The paper has a clear motivation, aiming to analyze the generalization bounds of two-stage tree-structured recommender systems. The writing is clear and understandable.
2. Two-stage recommender systems are indeed one of the mainstream structures in the current field of recommender systems, especially in industry. However, there has been a lack of theoretical guarantees and guidance for such systems. This paper effectively highlights the issues related to the number of branches in tree structures and the training data for rankers, which is innovative and high-quality.
3. The paper theoretically studies the relationship between the generalization of tree-structured Retrievers and the number of branches, as well as the relationship between the generalization of rankers and the difference between their training data distribution and the Retriever's predicted distribution. These conclusions are reasonable and align with empirical knowledge.
4. The experimental results nicely validate the theoretical findings, making the paper cohesive and solid.

**Weaknesses:**

1. The theoretical results regarding the generalization bounds of tree-structured Retrievers are highly similar to the results for hierarchical multi-class classification in reference [1], with the main difference being the consideration of the Beam Search algorithm. Additionally, the analysis of the Rademacher complexity for linear models and MLP models has been previously established. These should be referenced in the main text of the paper.
2. The paper mentions: "In our experiments, we found that a recall rate of more than 10% is typically required to see an improvement effect" in line 271.  Is there a corresponding experimental analysis for this conclusion? For example, how much does the performance of the ranker model improve with different recall rates (i.e., different data volumes)? Moreover, can this conclusion be theoretically justified as well?
3. There are some typos that need to be checked, especially in the use of bold and regular fonts. For instance, the subscript 'c' in the first formula on page 4 is not bolded, and the subscript 'v’' in formula 10 in Appendix A is not bolded.

**Questions:**

See weaknessnes.

**Limitations:**

NA.

---

> ### Author Rebuttal · Authors · 2024-08-05
>
> Thank you for your recognition of our work and providing the constructive comments. We will try our best to address your concerns with planned revisions based on your valuable feedback.
>
> **Response to Weakness 1:**
>
> Thank you for your detailed feedback. Our analysis of tree-structured models indeed draws inspiration from previous work, particularly in the context of hierarchical multi-class classification[1]. Our contribution, in addition to the main difference of extending the analysis to the Beam Search algorithm, also provides a more refined estimate specifically tailored to the tree model. Specifically, we use the mapping $c(f, \boldsymbol{x}, y)=\left(\boldsymbol{v}, \boldsymbol{v}^{\prime}\right)$, which establishes a correspondence between a sample point $(x,y)$ and both the target node and the Top-K scoring nodes along the search path within the tree. This mapping allows us to partition the $m$ sample points into several disjoint sets $ \lbrace (x_i,y_i): c\left(f, \boldsymbol{x}_i, y_i\right)=\left(\boldsymbol{v}, \boldsymbol{v}^{\prime}\right) \rbrace $, each corresponding to different node pairs in the tree. Unlike prior work[1],  the proof of Theorem 1 in [1], where a trivial upper bound of $m$ is used for the size of these sets, we provide individual estimates for each set’s size. By combining these estimates, we derive a tighter overall result, leading to a key conclusion that increasing the number of branches helps to reduce the generalization error—an insight not directly captured in [1].
>
> Regarding Rademacher complexity, our model introduces an additional tree structure, distinguishing it from traditional linear models and MLPs. Our analytical approach involves separating the traditional scoring model from the tree structure component, corresponding to the term $\mathcal{T}$ in the upper bound of the Rademacher complexity in our analysis. This reduction maps the problem to the scoring function space of traditional models, making existing analysis techniques applicable. We appreciate your comments regarding our references, and we will ensure that these works are properly cited in future revisions.
>
> [1] Rohit Babbar et al. Learning taxonomy adaptation in large-scale classification. JMLR, 17(1):3350–3386, 2016.
>
> **Response to Weakness 2:**
>
> Thank you for your observation. We will include experimental results in the revised manuscript to demonstrate the ranker model's performance at different recall rates.
>
> In our experiments, we varied the number of items retrieved by the model to adjust the recall rate. With $ K=40 $ fixed during inference, from Tables 1 and 2, it can be observed that the ranker model's performance significantly declined when the recall rate was as low as 7.5\% due to insufficient training data. However, when the recall rate exceeded 10.7\%, the model consistently showed improvement.
>
> The improvement in the ranker model's performance depends on sufficient training data and alignment with the target distribution.  It can be observed that once the recall rate reaches a sufficient threshold, further increases in the recall rate actually cause the performance of the trained ranker model in the two-stage classification process to gradually decline, approaching the performance of the ranker model trained on the original distribution, i.e., the complete dataset. This decline occurs because the training data distribution starts to deviate from the target distribution. The most optimal setting may be to keep the number of retrieved items consistent between training and inference, provided the recall rate is relatively high in this scenario.
> Experimental results show that limited data initially hinders performance, but as data and alignment improve, performance increases. However, excessive data leads to misalignment and a subsequent performance decline, consistent with theoretical predictions. For a quantitative result in theory, if we use a sampling method to estimate the error between distributions, defined as
> $err_{\mathcal{D}} := \mathbb{E} _{ (\boldsymbol{x}, y) \sim \mathcal{D}}|1-\frac{P'(\boldsymbol{x}, y)}{P(\boldsymbol{x}, y)} | $, we have
>
> $$
> err_{\mathcal{D}} \approx \frac{1}{m}\sum_{i=1}^m \mathbb{I}[y_i \notin \mathcal{B}(x_i)] + \left(\frac{1}{m^\prime} - \frac{1}{m}\right) \sum_{i=1}^m \mathbb{I}[y_i \in \mathcal{B}(x_i)] ,
> $$
> and if we aim to achieve an $ \epsilon $ error between the generalization error and the empirical error with probability $1-\delta $, the estimated number of training samples required can be expressed as:
>
> $$
> m \geq \left(\frac{4 c_{\Phi} N(K+1) B_{\text{model}} + B_{\Phi} \sqrt{2 \log (2 / \delta)}}{\epsilon - {err}_{\mathcal{D}}}\right)^2.
> $$
> By further comparing this estimate with the total number of samples, we can estimate the required recall rate. It is worth noting such an estimate is typically conservative, and we still recommend using the results from the experiments.
>
> **Table 1: Model Performance on the Mind Dataset**
>
> | Recall | Accuracy | Improvement (Above 0.6500) |
> | ----------- | -------- | ----------------------------------- |
> | 16.1%       | 0.6717   | Yes                                 |
> | 25.0%       | 0.6844   | Yes                                 |
> | 36.2%       | 0.6685   | Yes                                 |
> | 49.1%       | 0.6550   | Yes                                 |
>
> **Table 2: Model Performance on the Movie Dataset**
>
> | Recall | Accuracy | Improvement (Above 0.3516) |
> | ----------- | -------- | ----------------------------------- |
> | 7.5%        | 0.2581   | No                                  |
> | 10.7%       | 0.3548   | Yes                                 |
> | 17.4%       | 0.3562   | Yes                                 |
> | 24.6%       | 0.3547   | Yes                                 |
>
>
> **Response to Weakness 3:**
>
> Thank you for identifying the typos. We will correct these issues and ensure consistency in formatting throughout the manuscript.

---

### Official Review · Reviewer_pcvU · 2024-07-18

**Soundness:** 3
**Presentation:** 3
**Contribution:** 4
**Rating:** 8
**Confidence:** 4

**Summary:**

This paper analyzes the generalization error of two-stage recommender systems with a tree structure, which consist of an efficient tree-based retriever and a more precise but time-consuming ranker. The authors use Rademacher complexity to establish generalization error upper bounds for various tree-based retrievers using beam search, as well as for different ranker models under a shifted training distribution. The key findings are that increasing the number of branches in tree-based retrievers and harmonizing distributions across stages can enhance the overall generalization performance of two-stage recommender systems, as validated through both theoretical insights and practical experiments.

**Strengths:**

The paper provides a comprehensive theoretical analysis of the generalization error bounds for two-stage recommender systems with a tree structure. This is an important contribution, as previous theoretical research in this area has been limited.

The paper analyzes the generalization upper bounds for various tree-based retriever models using beam search, including linear models, multilayer perceptrons, and target attention models. This provides valuable insights into the learnability and generalization capabilities of these widely used retriever models.

The paper analyzes the generalization upper bounds for ranker models under shifted training distributions. This is an important consideration, as the data distribution during inference can often differ from the training distribution in real-world recommender systems. The theoretical and empirical findings on harmonizing distributions across stages are valuable insights.

The theoretical insights and guidelines derived in this paper can inform the design and development of more robust and generalizable two-stage recommender systems, with significant implications for a wide range of industries and applications. The analytical techniques and the established error decomposition framework can serve as a foundation for future research in this domain.

**Weaknesses:**

The analysis of tree-based retriever models is comprehensive, but the paper does not explore the generalization properties of other types of retriever architectures, such as deep learning-based models beyond the target attention model. Expanding the analysis to a broader range of retriever models could provide a more holistic understanding of two-stage recommender systems.

The paper primarily focuses on the generalization performance, but does not delve into the computational complexity and inference latency of the proposed two-stage recommender systems. Providing a more comprehensive analysis of the efficiency and scalability aspects would further strengthen the practical relevance of the work.

**Questions:**

The analysis of tree-based retriever models is comprehensive, but does the paper explore the generalization properties of other types of retriever architectures, such as deep learning-based models beyond the target attention model? Would expanding the analysis to a broader range of retriever models provide a more holistic understanding of two-stage recommender systems?

While the paper primarily focuses on the generalization performance, does it delve into the computational complexity and inference latency of the proposed two-stage recommender systems? Would providing a more comprehensive analysis of the efficiency and scalability aspects further strengthen the practical relevance of the work?

---

> ### Author Rebuttal · Authors · 2024-08-05
>
> We appreciate your recognition of our work and the opportunity to address the concerns raised in your review. We value your insightful feedback and would like to share our thoughts in response.
>
> **Response to Questions 1:**
>
>
> > **Questions 1:** The analysis of tree-based retriever models is comprehensive, but the paper does not explore the generalization properties of other types of retriever architectures, such as deep learning-based models beyond the target attention model. Expanding the analysis to a broader range of retriever models could provide a more holistic understanding of two-stage recommender systems?
>
> Thanks for your question. We fully agree that exploring retriever architectures beyond tree-based models could provide a more comprehensive understanding of two-stage recommender systems. This is a promising and broad area of research. In this work, we chose to begin with an analysis of the generalization bounds for tree-based retriever models, which are commonly used in current two-stage models, with the hope of contributing to research in this broader area. We will continue to work in this direction to advance and refine research in this area in the future.
>
> **Response to Questions 2:**
>
>
> > **Questions 2:** While the paper primarily focuses on the generalization performance, does it delve into the computational complexity and inference latency of the proposed two-stage recommender systems? Would providing a more comprehensive analysis of the efficiency and scalability aspects further strengthen the practical relevance of the work?
>
> Thanks again for your question. We also share the same view that efficiency and scalability are key considerations in practical recommender systems.
>
> In terms of efficiency, specifically computational complexity and inference latency, which are critical concerns in practical recommender systems, our work points to the impact of the number of branches on the performance of tree-based models. While the generalization-related conclusions may not directly correlate with efficiency, we highlight that increasing the number of branches in the tree can improve model performance, but also increases the computational complexity and inference latency of the retriever. In an extreme case, when the number of branches equals the number of items, the tree structure becomes ineffective because it requires traversing all items during inference. At this point, the retriever model essentially degenerates into a ranker model, which is more precise yet more time-consuming. The number of branches can thus be viewed as a tradeoff between performance and efficiency.
>
> In terms of scalability, the two improvement strategies derived from our theoretical analysis, increasing the number of tree branches and adjusting the training distribution of the ranker, are feasible in practice across different models. Although the conclusions may vary depending on the specific network architecture, these strategies can still provide valuable guidance for model design.

---

### Decision · Program_Chairs · 2024-09-25

**Decision:**

Accept (oral)

**Comment:**

The paper considers two-stage recommender systems with tree structure. The theoretical work, providing generalization error of the retriever and ranker components, were appreciated by all reviewers. While all reviewers recommended acceptance, some issues were raised (and addressed) in the reviews. Please make sure to incorporate those changes into the final version of the paper.